# Convergent Differential Privacy Analysis for General Federated Learning

## Abstract

The powerful cooperation of federated learning (FL) and differential privacy (DP) provides a promising paradigm for the large-scale private clients. However, existing analyses in FL-DP mostly rely on the composition theorem and cannot tightly quantify the privacy leakage challenges, which is tight for a few communication rounds but yields an arbitrarily loose and divergent bound eventually. This also implies a counterintuitive judgment, suggesting that FL-DP may not provide adequate privacy support during long-term training under constant-level noisy perturbations, yielding discrepancy between the theoretical and experimental results. To further investigate the convergent privacy and reliability of the FL-DP framework, in this paper, we comprehensively evaluate the worst privacy of two classical methods under the non-convex and smooth objectives based on the $f$-DP analysis. With the aid of the shifted interpolation technique, we successfully prove that privacy in `Noisy-FedAvg` has a tight convergent bound. Moreover, with the regularization of the proxy term, privacy in `Noisy-FedProx` has a stable constant lower bound. Our analysis further demonstrates a solid theoretical foundation for the reliability of privacy in FL-DP. Meanwhile, our conclusions can also be losslessly converted to other classical DP analytical frameworks, e.g. $(\epsilon, \delta)$-DP and Rényi-DP (RDP), to provide more fine-grained understandings for the FL-DP frameworks.

## 1 Introduction

Since McMahan et al. [2017] proposes the `FedAvg` method as a general FL framework, it has been widely developed into a collaborative training standard with privacy protection attributes, which successfully avoids *direct leakage* of sensitive data. As research on privacy progresses, researchers have found that standard FL frameworks still face a threat from *indirect leakage*. Attackers can potentially recover local private data through reverse inference by persistently stealing model states via model (gradient) inversion attacks [Geiping et al., 2020] or distinguish whether individuals are involved in the training via membership inference attacks [Nasr et al., 2019]. To further strengthen the reliability of FL, DP [Dwork, 2006, Dwork et al., 2014, Abadi et al., 2016] has naturally been incorporated into the FL framework, yielding FL-DP [Wei et al., 2020]. As a primary technique, the noisy perturbation is widely applied in various advanced FL methods to further enhance its security.

However, the theoretical analysis of the FL-DP framework, especially in evaluating the privacy levels, is currently unable to provide a comprehensive understanding of its proper application. Most of the previous works are built upon the foundational lemma of privacy amplification by iteration, directly resulting in divergent privacy bound as the training communication round $T$ becomes large. This implies an inference that contradicts intuition and empirical studies, which is, that the FL-DP framework may completely lose its privacy protection attributes as $T \rightarrow \infty$. Such a conclusion is almost unacceptable for FL-DP. Therefore, establishing a precise and tight analysis is a crucial target.

Submitted to 39th Conference on Neural Information Processing Systems (NeurIPS 2025). Do not distribute.

Table 1: The worst privacy of the `Noisy-FedAvg` and `Noisy-FedProx` methods in our analysis. $V$ is the norm of clip gradient. $K, T$ are local training interval and communication round. $\sigma$ is the variance of the noise. The trade-off function $T_G(\cdot)$ [a] is defined in Definition 4. $\mu$, $c$ and $z$ are constants.

| | Lr [b] | Worst Privacy | Convergent? on $T \to \infty$ | Convergent? on $K \to \infty$ |
|---|---|---|---|---|
| Noisy FedAvg | C | $T_G\left(\frac{2\mu V K}{\sqrt{m}\sigma}\sqrt{\frac{(1+\mu L)^K+1}{(1+\mu L)^K-1}\frac{(1+\mu L)^{KT}-1}{(1+\mu L)^{KT}+1}}\right)$ | ✔ | ✘ |
| | CD | $T_G\left(\frac{2cV\ln(K+1)}{\sqrt{m}\sigma}\sqrt{\frac{(1+K)^{c\mu L}+1}{(1+K)^{c\mu L}-1}\frac{(1+K)^{c\mu LT}-1}{(1+K)^{c\mu LT}+1}}\right)$ | | |
| | SD | $T_G\left(\frac{2\mu V K}{\sqrt{m}\sigma}\sqrt{2-\frac{1}{T}}\right)$ | | |
| | ID | $T_G\left(\frac{2zV}{\sqrt{m}\sigma}\sqrt{2-\frac{1}{T}}\right)$ | ✔ | ✔ |
| Noisy FedProx | | $T_G\left(\frac{2V}{\sqrt{m}\alpha\sigma}\sqrt{\frac{2\alpha-L}{L}}\sqrt{\frac{\alpha^T-(\alpha-L)^T}{\alpha^T+(\alpha-L)^T}}\right)$ | | |

[a] For the trade-off function $T_G(s)$, smaller $s$ means stronger privacy.
[b] Learning rate decaying policy. C: constant learning rate; CD: cyclically decaying; SD: stage-wise decaying; ID: iteratively decaying. More details are stated in Theorem 3 4.

Notably, significant progress has been made in characterizing convergent privacy in the noisy gradient descent method in RDP analysis [Chourasia et al., 2021, Ye and Shokri, 2022, Altschuler and Talwar, 2022, Altschuler et al., 2024]. However, due to the challenges and intricacies of the analytical techniques adopted, similar results have not yet successfully been extended to the FL-DP. The multi-step local updates on heterogeneous datasets lead to biased local models, posing significant obstacles to the analysis. Recently, analyses based on $f$-DP [Dong et al., 2022] have brought a promising resolution to this challenge. This information-theoretically lossless definition naturally evaluates privacy by the Type I / II error trade-off curve of the hypothesis testing problem about whether a given individual is in the training dataset. Combined with shifted interpolation techniques [Bok et al., 2024], it successfully recovers tighter convergent privacy for strongly convex and convex objectives in noisy gradient descent methods. This may make it possible to quantify convergent privacy in FL-DP and may offer novel understandings about impacts of some key hyperparameters.

In this paper, we investigate the privacy of two classic DP-FL methods, i.e. `Noisy-FedAvg` and `Noisy-FedProx` and successfully evaluate their *worst privacy* in the $f$-DP analysis, as shown in Table 1. For the `Noisy-FedAvg` method, we investigate four typical learning rate decay strategies and provide the coefficients corresponding to each case to ensure a tighter privacy lower bound. We also prove that its iterative privacy on non-convex and smooth objectives could not diverge w.r.t. the number of communication rounds $T$, i.e., a convergent privacy. To the best of our knowledge, this contributes the first convergent privacy analysis in FL-DP methods for non-convex functions. Furthermore, by exploring the decay properties of the proximal term in `Noisy-FedProx`, we prove that its worst privacy can converge to a general constant lower bound. Our analysis successfully challenges the long-standing belief that privacy budgets of FL-DP have to increase as training processes and provides reliable guarantees for its privacy protection ability. At the same time, the exploration from the proximal term provides a promising solution, suggesting that a well-designed local regularization term can achieve a win-win solution for both optimization and privacy in FL-DP.

## 2  Related Work

**Federated Learning.** FL is a classic learning paradigm that protects local privacy. Since McMahan et al. [2017] proposes the basic framework, it has been widely studied in several communities. As its foundational study, the `local-SGD` [Stich, 2019, Lin et al., 2019, Woodworth et al., 2020, Gorbunov et al., 2021] method fully demonstrates the efficiency of local training. Based on this, FL further considers the impacts of heterogeneous private datasets and communication bottlenecks [Wang et al., 2020, Chen et al., 2021, Kairouz et al., 2021]. To address these two basic issues, a series of studies have explored these processes from different perspectives. One approach involves proposing better

optimization algorithms by defining concepts such as client drift [Karimireddy et al., 2020] and heterogeneity similarity [Mendieta et al., 2022], specifically targeting and resolving the additional error terms they cause. This mainly includes the natural application and expansion of variance-reduction optimizers [Jhunjhunwala et al., 2022, Malinovsky et al., 2022, Li et al., 2023], the flexible implementation of the advanced primal-dual methods [Zhang et al., 2021c, Wang et al., 2022, Sun et al., 2023b, Mishchenko et al., 2022, Grudzień et al., 2023, Acar et al., Sun et al., 2023a], and the additional deployment of the momentum-based correction [Liu et al., 2020, Khanduri et al., 2021, Das et al., 2022, Sun et al., 2023c, 2024]. Upgraded optimizers allow the aggregation frequency to largely decrease while maintaining convergence. Another approach primarily focuses on sparse training and quantization to reduce communication bits [Reisizadeh et al., 2020, Shlezinger et al., 2020, Dai et al., 2022]. Additionally, research based on data domain and feature domain has also made significant contributions to the FL community [Yao et al., 2019, Zhang et al., 2021a, Xu et al.].

**FL-DP.** DP is a natural privacy-preserving framework with theoretical foundations [Dwork et al., 2006b,a, Dwork, 2006]. As one of the main algorithms for differential privacy, noise perturbation has achieved great success in deep learning [Abadi et al., 2016, Zhao et al., 2019, Arachchige et al., 2019, Wu et al., 2020]. Combining this, FL-DP adds noise before transmitting their variables, i.e. client-level noises [Geyer et al., 2017] and server-level noises [Wei et al., 2020]. Since there is no fundamental difference between the analysis of them, in this paper, we mainly consider client-level noises. One major research direction involves conducting noise testing on widely developed federated optimization algorithms [Zhu et al., 2021, Noble et al., 2022, Lowy et al., 2023, Zhang and Tang, 2022, Yang and Wu, 2023], and evaluating the performance of different methods under DP noises through convergence analysis and privacy analysis. Another research direction involves injecting noise into real-world systems to address practical challenges, which primarily focuses on personalized scenarios [Hu et al., 2020, Yang et al., 2021, 2023, Wei et al., 2023], decentralized scenarios [Wittkopp and Acker, 2020, Chen et al., 2022, Gao et al., 2023, Shi et al., 2023], and adaptive or asymmetric update scenarios [Girgis et al., 2021, Wu et al., 2022, He et al., 2023]. FL-DP has been extensively tested across various scales of tasks and has successfully validated its robust local privacy protection capabilities. At the same time, the theoretical analysis of FL-DP has been progressing systematically and in tandem. Based on various DP relaxations, they provide a comparison of privacy performance by analyzing concepts such as privacy budgets, and further understand the specific attributes of privacy algorithms [Rodríguez-Barroso et al., 2020, Wei et al., 2021, Kim et al., 2021, Zheng et al., 2021, Ling et al., 2024, Jiao et al., 2024]. Theoretical advancements in DP have revolutionized how we could quantify and safeguard privacy, offering unprecedented precision and robustness.

# 3 Preliminaries

**Notations.** In the subsequent content, we use italics for scalars and denote the integer set from 1 to $a$ by $[a]$. All sequences of variables are represented in subscript, e.g. $w_{i,k,t}$. For arithmetic operators, unless specifically stated otherwise, the calculations are performed element-wise. Other symbols used in this paper will be explicitly defined when they are first introduced.

## 3.1 General FL-DP framework

We consider the general finite-sum minimization problem in the classical federated learning:

$$w^\star \in \arg\min_w f(w) \triangleq \frac{1}{m} \sum_{i \in \mathcal{I}} f_i(w), \tag{1}$$

where $f_i(w) = \mathbb{E}_{\varepsilon \sim \mathcal{D}_i}\left[f_i(w, \varepsilon)\right]$ denotes the local population risk. $w \in \mathbb{R}^d$ denotes $d$-dim learnable parameters. $\varepsilon \sim \mathcal{D}_i$ denotes that the private dataset on client $i$ is sampled from distribution $\mathcal{D}_i$. We consider the general heterogeneity, i.e. $\mathcal{D}_i$ can differ from $\mathcal{D}_j$ if $i \neq j$, leading to $f_i(w) \neq f_j(w)$.

In our analysis, we consider the FL-DP framework with the classical client-level Gaussian noises. The FL training process remains consistent with standard training procedures. The local clients enhance local privacy by adding isotropic Gaussian noises to the uploaded model parameters, i.e. $n_i \sim \mathcal{N}(0, \sigma^2 I_d)$. Then the global server aggregates the noisy parameters as the global model $w_{t+1}$. Due to the page limitation, details of the algorithmic implementation are deferred to the Appendix A.

**Noisy-FedAvg:** we consider that each local client performs a fundamental gradient descent as follows:

$$w_{i,k+1,t} = w_{i,k,t} - \eta_{k,t} g_{i,k,t}, \tag{2}$$

where $g_{i,k,t} = \nabla f_i(w_{i,k,t}, \varepsilon) / \max\{1, \frac{\|\nabla f_i(w_{i,k,t}, \varepsilon)\|}{V}\}$, and $V$ is a constant coefficient.

**Noisy-FedProx:** The vanilla local training in `FedProx` is based on solving the following surrogate:

$$\min_w f_i(w) + \frac{\alpha}{2}\|w - w_t\|^2. \tag{3}$$

To generally compare with `Noisy-FedAvg`, we consider an iterative form of gradient descent as:

$$w_{i,k+1,t} = w_{i,k,t} - \eta_{k,t}\left[g_{i,k,t} + \alpha(w_{i,k,t} - w_t)\right]. \tag{4}$$

### 3.2 DP and $f$-DP

**Definition 1** *We denote heterogeneous datasets on the client $i$ by $\mathcal{S}_i = \{\varepsilon_{ij}\}$ and let the union of all local datasets be $\mathcal{C} = \{\mathcal{S}_i\}$. We say two unions are adjacent datasets if they only differ by one data sample. For instance, there exists the union $\mathcal{C}' = \{\mathcal{S}_i'\}$. $(\mathcal{C}, \mathcal{C}')$ are adjacent datasets if there exists the index pair $(i^\star, j^\star)$ such that all other data samples are the same except for $\varepsilon_{i^\star j^\star} \neq \varepsilon_{i^\star j^\star}'$.*

**Definition 2** *A randomized mechanism $\mathcal{M}$ is $(\epsilon, \delta)$-DP if for any event $E$ the following satisfies:*

$$P(\mathcal{M}(\mathcal{C}) \in E) \leq e^\epsilon P(\mathcal{M}(\mathcal{C}') \in E) + \delta. \tag{5}$$

Definition 2 is the widely used $(\epsilon, \delta)$-DP, which is a lossy relaxation in the DP analysis since its probabilistic gaps. To bridge the discrepancy of precise DP definitions, statistic analysis demonstrates that DP could be naturally deduced by hypothesis-testing problems [Wasserman and Zhou, 2010, Kairouz et al., 2015]. From the perspective of attackers, DP means the difficulty in distinguishing $\mathcal{C}$ and $\mathcal{C}'$ under the mechanism $\mathcal{M}$. They can generally consider the following problem:

*Given $\mathcal{M}$, is the underlying union $\mathcal{C}$ ($H_0$) or $\mathcal{C}'$ ($H_1$)?*

To exactly quantify the difficulty of its answer, Dong et al. [2022] propose that distinguishing these two hypotheses could be best delineated by the optimal trade-off between the possible type I and type II errors. Specifically, by considering rejection rules $0 \leq \chi \leq 1$, type I and type II errors can be:

$$E_I = \mathbb{E}_{\mathcal{M}(\mathcal{C})}[\chi], \qquad E_{II} = 1 - \mathbb{E}_{\mathcal{M}(\mathcal{C}')}[\chi], \tag{6}$$

Here, we abuse $\mathcal{M}(\mathcal{C})$ to represent its probability distribution. To measure the fine-grained relationships between these two testing errors, $f$-DP is introduced.

**Definition 3 (Trade-off function)** *For any two probability distributions $P$ and $Q$, the trade-off function is defined as: $T(P; Q)(\gamma) = \inf\{1 - \mathbb{E}_Q[\chi] \mid \mathbb{E}_P[\chi] \leq \gamma\}$, where the infimum is taken over all measurable rejection rules.*

$T(P; Q)(\gamma)$ is convex, continuous, and non-increasing. For any possible rejection rules, it satisfies $T(P; Q)(\gamma) \leq 1 - \gamma$. It functions as the clear boundary between the achievable and unachievable selections of type I and type II errors, essentially distinguishing the difficulties between these two hypotheses. This relevant statistical property provides a stricter definition of privacy, which mitigates the excessive relaxation of privacy based on composition analysis in existing approaches.

**Definition 4 ($f$-DP and GDP)** *A mechanism $\mathcal{M}$ is $f$-DP if $T(\mathcal{M}(\mathcal{C}), \mathcal{M}(\mathcal{C}'))(\gamma) \geq f(\gamma)$ for all possible adjacent datasets $\mathcal{C}$ and $\mathcal{C}'$. When $f$ measures two Gaussian distributions, namely Gaussian-DP (GDP), denoted as $T_G(\mu)(\gamma) \triangleq T(\mathcal{N}(0, 1), \mathcal{N}(\mu, 1))(\gamma)$ for $\mu \geq 0$.*

According to the definition, the explicit representation of GDP is $T_G(\mu)(\gamma) = \Phi(\Phi^{-1}(1 - \gamma) - \mu)$ where $\Phi$ denotes the standard Gaussian CDF. Any single sampling mechanism that introduces Gaussian noises can be considered as an exact GDP, which monotonically decreases when $\mu$ increases.

## 4 Convergent Privacy

In this section, we primarily demonstrate how to provide the worst privacy in FL-DP and its convergent bound. Generally, we assume that local objectives satisfy smoothness with a constant $L$,

**Assumption 1** *Each local objective function $f_i(\cdot)$ satisfies $L$-smoothness, i.e.,*

$$\|\nabla f_i(w_1) - \nabla f_i(w_2)\| \leq L\|w_1 - w_2\|. \tag{7}$$

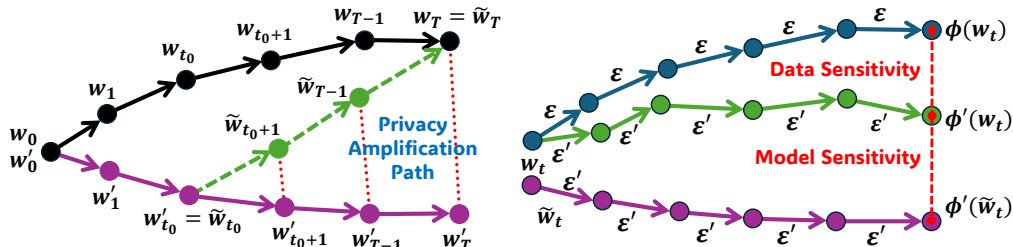

Figure 1: *Left*: The global privacy amplification path induced by the shifted interpolation sequence. *Right*: Estimation of the global sensitivity under local updates via an auxiliary sequence.

### 4.1 Shifted Interpolation

To simplify presentations, we denote global updates at round $t$ on the adjacent datasets $\mathcal{C}$ and $\mathcal{C}'$ as:

$$\mathcal{C} : w_{t+1} = \phi(w_t) + \overline{n}_t, \quad \mathcal{C}' : w'_{t+1} = \phi'(w'_t) + \overline{n}'_t. \tag{8}$$

$\phi(w_t)$ denotes the accumulation of total $K$ steps from the initialization state $w_{i,0,t} = w_t$ at round $t$. $\overline{n}_t$ could be considered as the averaged noise, i.e. $\overline{n}_t \sim \mathcal{N}(0, \sigma^2 I_d/m)$. Traditional methods require performing privacy amplification $T$ times based on the relationship between $w$ and $w'$, yielding non-convergent privacy as $T$. To avoid loose privacy amplification, we follow Bok et al. [2024] to adopt the *shifted interpolation* technique. Specifically, we define the following sequence:

$$\widetilde{w}_{t+1} = \lambda_{t+1}\phi(w_t) + (1 - \lambda_{t+1})\phi'(\widetilde{w}_t) + \overline{n}_t, \tag{9}$$

where $t = t_0, \cdots, T-1$. By setting $\lambda_T = 1$, then $\widetilde{w}_T = w_T$, and we add the definition of $\widetilde{w}_{t_0} = w'_{t_0}$ as the beginning of interpolations. $0 \le \lambda_t \le 1$ are interpolation coefficients to be optimized. As shown in Figure 1 (left), the interpolation sequence path enables a privacy amplification analysis over $T - t_0$ times where $t_0$ is an optimizable coefficient. Therefore, we can establish the following theorem along this new privacy amplification path.

**Theorem 1** *Under Assumption 1 and corresponding updates in Eq.(8), After $T$ training rounds on the adjacent datasets $\mathcal{C}$ and $\mathcal{C}'$, we can bound the trade-off function between $w_T$ and $w'_T$ as:*

$$T(w_T; w'_T) = T(\widetilde{w}_T; w'_T) \ge T_G \left( \frac{\sqrt{m}}{\sigma} \sqrt{\sum_{t=t_0}^{T-1} \lambda_{t+1}^2 \|\phi(w_t) - \phi'(\widetilde{w}_t)\|^2} \right). \tag{10}$$

In addition to the influence of standard parameters, Theorem 1 highlights the critical relationship between the privacy lower bound and the weighted sum of global sensitivity terms from $t_0$ to $T$. Therefore, we then analyze the global sensitivity term $\|\phi(w_t) - \phi'(\widetilde{w}_t)\|$.

### 4.2 Global Sensitivity

The sensitivity term $\|\phi(w_t) - \phi'(\widetilde{w}_t)\|^2$ means the stability gaps between $w_t$ and $\widetilde{w}_t$ after performing local training on datasets $\mathcal{C}$ and $\mathcal{C}'$ respectively. It is influenced by both the model parameters and the data samples, making the analysis extremely challenging. To achieve a fine-grained analysis, we propose an auxiliary sequence $\phi'(w_t)$. As shown in Figure 1 (right), the global sensitivity can be split into *data sensitivity* and *model sensitivity*. The *data sensitivity* measures the estimable errors obtained after training on different datasets for several steps from the same initialization. This discrepancy is solely caused by the data. The *model sensitivity* measures the estimable errors of the updates when two different initialized states are trained on the same dataset. Clearly, this discrepancy is directly related to the degree of similarity between the two initializations. Thus, we have:

**Theorem 2** *Under $K$ local updates by Eq.(2) and Eq.(4), the global sensitivity in `Noisy-FedAvg` and `Noisy-FedProx` methods can be shown as:*

$$\|\phi(w_t) - \phi'(\widetilde{w}_t)\| \le \underbrace{\rho_t \|w_t - \widetilde{w}_t\|}_{\text{from model sensitivity}} + \underbrace{\gamma_t}_{\text{from data sensitivity}}, \tag{11}$$

*where $\rho_t$ and $\gamma_t$ are shown in Table 2.*

Table 2: Specific formulation of $\rho_t$ and $\gamma_t$ in Theorem 2.

| | Learning rate | $\rho_t$ | $\gamma_t$ |
|---|---|---|---|
| `Noisy-FedAvg` | $\mu$ | $(1 + \mu L)^K$ | $\frac{2\mu V}{m} K$ |
| | $\frac{\mu}{k+1}$ | $(1 + K)^{c\mu L}$ | $\frac{2cV}{m} \ln(K+1)$ |
| | $\frac{\mu}{t+1}$ | $\left(1 + \frac{\mu L}{t+1}\right)^K$ | $\frac{2\mu V}{m} \frac{K}{t+1}$ |
| | $\frac{\mu}{tK+k+1}$ | $\left(\frac{t+2}{t+1}\right)^{z\mu L}$ | $\frac{2zV}{m} \ln\left(\frac{t+2}{t+1}\right)$ |
| `Noisy-FedProx` | non-increase | $\frac{\alpha}{\alpha-L}$ | $\frac{2V}{m\alpha}$ |

**Remark 2.1** *The result in Eq.(11) aligns with the intuition of designing the splitting operators. It can be observed that the coefficient $\rho_t$ is consistently greater than 1, which is a typical characteristic of non-convexity. It also implies that the sensitivity upper bound tends to diverge as $t \to \infty$. However, in Eq.(10), the parameters $0 \le \lambda_t \le 1$ can efficiently scale the sensitivity terms. By carefully selecting the optimal $\lambda_t$ values, it can ultimately achieve a convergent privacy lower bound.*

### 4.3 Minimization Problem on $t_0$ and Its Relaxation

According to Eq.(10) and the sensitivity bound in Eq.(11), we denote the weighted accumulation of the sensitivity term as $\mathcal{H}(\lambda_t, t_0)$, where $\lambda_t$ and $t_0$ are both to-be-optimized parameters. Therefore, we can provide the tight bound of the privacy by solving the minimization of the following problem:

$$\mathcal{H}_\star = \min_{\lambda_t, t_0} \mathcal{H}(\lambda_t, t_0) \triangleq \sum_{t=t_0}^{T-1} \lambda_{t+1}^2 \left(\rho_t \|w_t - \widetilde{w}_t\| + \gamma_t\right)^2. \tag{12}$$

If $t_0$ is very small, it means that the introduced stability gap will also be very small. However, consequently, the sensitivity terms will extremely increase due to the accumulation over $T - t_0$ rounds. Conversely, although the accumulated error is small, it remains divergent due to the unbounded global sensitivity term. To avoid this uncertain analysis, we have to make a compromise. Because $t_0$ is an integer belonging to $[0, T-1]$, its optimal selection certainly exists when $T$ is given. Therefore, we consider a relaxed and simple problem instead, i.e. under $t_0 = 0$,

$$\mathcal{H}_0 = \min_{\lambda_t} \mathcal{H}(\lambda_t, 0) = \sum_{t=0}^{T-1} \lambda_{t+1}^2 \left(\rho_t \|w_t - \widetilde{w}_t\| + \gamma_t\right)^2. \tag{13}$$

Its advantage lies in the fact that when $t_0 = 0$, the sensitivity error is 0, avoiding its divergence. Compared to the optimal solution $\mathcal{H}_\star$, it satisfies $\mathcal{H}_0 \ge \mathcal{H}_\star$. More importantly, the solution of $\mathcal{H}_0$ eliminates the influence of $t_0$, allowing us to obtain an effective solution to the minimization problem by directly minimizing the $\lambda_t$ terms. The lower bound in Theorem 1 will be replaced by:

$$T(w_T; w_T') \ge T_G\left(\frac{\sqrt{m\mathcal{H}_\star}}{\sigma}\right) \ge T_G\left(\frac{\sqrt{m\mathcal{H}_0}}{\sigma}\right). \tag{14}$$

Although this is a relaxation of the privacy lower bound, our subsequent proof confirms that $\mathcal{H}_0$ can still achieve convergent into a constant form, which means local privacy can still achieve convergence.

### 4.4 Convergent Privacy

In this part, we demonstrate our convergent privacy analysis. By solving Eq.(13) under corresponding $\rho_t$ and $\gamma_t$, we provide the worst privacy for the `Noisy-FedAvg` and `Noisy-FedProx` methods.

**Theorem 3** *Let $f_i(w)$ be a L-smooth and non-convex local objective and local updates be performed as shown in Eq.(2). Under perturbations of isotropic noises $n_i \sim \mathcal{N}\left(0, \sigma^2 I_d\right)$, the worst privacy of the* `Noisy-FedAvg` *method achieves:*

*(a) under constant learning rates $\eta_{k,t} = \mu$:*

$$T(w_T; w_T') \ge T_G\left(\frac{2\mu V K}{\sqrt{m}\sigma}\sqrt{\frac{(1+\mu L)^K + 1}{(1+\mu L)^K - 1}\frac{(1+\mu L)^{KT} - 1}{(1+\mu L)^{KT} + 1}}\right). \tag{15}$$

(b) under cyclically decaying $\eta_{k,t} = \frac{\mu}{k+1}$:

$$T(w_T; w_T') \geq T_G \left( \frac{2cV \ln(K+1)}{\sqrt{m}\sigma} \sqrt{\frac{(1+K)^{c\mu L}+1}{(1+K)^{c\mu L}-1} \frac{(1+K)^{c\mu LT}-1}{(1+K)^{c\mu LT}+1}} \right). \tag{16}$$

(c) under stage-wise decaying $\eta_{k,t} = \frac{\mu}{t+1}$:

$$T(w_T; w_T') > T_G \left( \frac{2\mu VK}{\sqrt{m}\sigma} \sqrt{2 - \frac{1}{T}} \right). \tag{17}$$

(d) under continuously decaying $\eta_{k,t} = \frac{\mu}{tK+k+1}$:

$$T(w_T; w_T') > T_G \left( \frac{2zV}{\sqrt{m}\sigma} \sqrt{2 - \frac{1}{T}} \right). \tag{18}$$

**Remark 3.1** *Theorem 3 provides the worst-case privacy analysis for the* `Noisy-FedAvg` *method. Its privacy is primarily affected by the clipping norm $V$, the local interval $K$, the scale $m$, and the noise intensity $\sigma$. A larger gradient clipping norm $V$ always results in larger gaps. The local interval $K$ determines the sensitivity of the entire local process, which is primarily influenced by the learning rate strategy. $m$ in our proof represents the client scale; in fact, the number of data samples is also proportional to $m$. An increased $m$ will largely reduce the sensitivity, yielding improvements in privacy. The impact of noise intensity $\sigma$ is also very intuitive. Infinite noise can provide perfect privacy, while zero noise offers no privacy. Constant-level noise can still achieve convergent privacy.*

**Theorem 4** *Let $f_i(w)$ be a $L$-smooth and non-convex local objective and local updates be performed as shown in Eq.(4). Let the proximal coefficient $\alpha > L$ and $\eta < \frac{1}{\alpha - L}$, under perturbations of isotropic noises $n_i \sim \mathcal{N}\left(0, \sigma^2 I_d\right)$, the worst privacy of the* `Noisy-FedProx` *method achieves:*

$$T(w_T; w_T') \geq T_G \left( \frac{2V}{\sqrt{m}\alpha\sigma} \sqrt{\frac{2\alpha - L}{L} \left( 1 - \frac{2}{\left(\frac{\alpha}{\alpha - L}\right)^T + 1} \right)} \right), \tag{19}$$

**Remark 4.1** *Aside from the influence of standard coefficients, due to the correction of the regularization term, its privacy is no longer affected by the local interval $K$, even with a constant learning rate, which becomes a significant advantage of the* `Noisy-FedProx` *method. Specifically, when $\alpha > L$, increasing $\alpha$ significantly improves the worst privacy, which achieves $\mathcal{O}\left(\frac{V}{\sigma\sqrt{m\alpha L}}\right)$ distinguishability in GDP. Therefore, the selection of $\alpha$ is a delicate trade-off between optimization and privacy. By selecting a proper $\alpha > L$, it enables a win-win outcome for both optimization and privacy.*

**Theoretical comparisons.** Table 3 demonstrates the comparison between existing theoretical results and ours of the `Noisy-FedAvg` method. Existing analyses are mostly based on the DP relaxations of $(\epsilon, \delta)$-DP and RDP [Mironov, 2017]. Apart from the lossiness in their DP definition, an important weakness is that privacy amplification on composition is entirely loose. For instance, the general amplification in $(\epsilon, \delta)$-DP indicates, the composition of an $(\epsilon_1, \delta_1)$-DP and an $(\epsilon_2, \delta_2)$-DP leads to an $(\epsilon_1 + \epsilon_2, \delta_1 + \delta_2)$-DP. Similarly, the composition of a $(\zeta, \epsilon_1)$-RDP and a $(\zeta, \epsilon_2)$-RDP results in a $(\zeta, \epsilon_1 + \epsilon_2)$-RDP. This simple parameter addition mechanism directly leads to a linear amplification of the privacy budget. Therefore, in previous works, when achieving specific DP guarantees, it is often required that the noise intensity $\sigma^2$ is proportional to the communication rounds $T$ (or $TK$). Wei et al. [2020] prove a double-noisy single-step local training on both client and server sides is possible to achieve the privacy amplification of $\mathcal{O}(T^2)$ rate. Shi et al. [2021] further consider the local intervals $K$. Zhang et al. [2021b] and Noble et al. [2022] elevate the theoretical results to $\mathcal{O}(TK)$. Subsequent research further indicates that the impact of the interval $K$ can be eliminated to achieve $\mathcal{O}(T)$ rate via sparsified perturbation [Hu et al., 2023, Cheng et al., 2022], and algorithmic improvements [Fukami et al., 2024]. However, these conclusions all indicate that the condition for achieving constant privacy guarantees is to continually increase the noise intensity. Bastianello et al. [2024] provide constant privacy under $\beta$-strongly convex objectives.

Table 3: Comparisons with the existing theoretical results in FL-DP. We losslessly transfer our results into $(\epsilon,\delta)$-DP and RDP results. In $(\epsilon,\delta)$-DP, we compare the requirement of noise variance corresponding to achieving $(\epsilon,\delta)$-DP. In $(\zeta,\epsilon)$-RDP, we directly compare the privacy budget term $\delta(\zeta)$. We mainly focus on the privacy changes on $T$ and $K$. $\Omega(\cdot)$, $\mathcal{O}(\cdot)$, and $o(\cdot)$ correspond to the lower, upper bound, and not tight upper bound of the complexity, respectively.

| | $(\epsilon,\delta)$-DP | $(\zeta,\epsilon)$-RDP | when $T,K\to\infty$ |
|---|---|---|---|
| Wei et al. [2020] | $\sigma = \mathcal{O}\left(\frac{V}{\epsilon m}\sqrt{T^2-mL^2}\right)$ | - | |
| Shi et al. [2021] | $\sigma = \mathcal{O}\left(\frac{V\sqrt{\log\left(\frac{1}{\delta}\right)}}{\epsilon}T\sqrt{K}\right)$ | - | |
| Zhang et al. [2021b] | $\sigma = \mathcal{O}\left(\frac{V\sqrt{\log\left(\frac{1}{\delta}\right)}}{\epsilon m}\sqrt{T+mK}\right)$ | - | |
| Noble et al. [2022] | $\sigma = \Omega\left(\frac{V\sqrt{\log\left(\frac{2T}{\delta}\right)}}{\epsilon\sqrt{m}}\sqrt{TK}\right)$ | - | $\sigma\to\infty$ on non-convex |
| Cheng et al. [2022] | $\sigma = \Omega\left(\frac{V\sqrt{\log\left(\frac{1}{\delta}\right)}}{\epsilon}\sqrt{T}\right)$ | - | |
| Zhang and Tang [2022] | - | $\epsilon = \Omega\left(\frac{\zeta V^2}{\sigma^2}TK\right)$ | |
| Hu et al. [2023] | $\sigma = \Omega\left(\frac{V\sqrt{\epsilon+2\log\left(\frac{1}{\delta}\right)}}{\epsilon}\sqrt{T}\right)$ | - | |
| Fukami et al. [2024] | $\sigma = \Omega\left(\frac{V(1+\sqrt{1+\epsilon})\sqrt{\log\left(e+\frac{\epsilon}{\delta}\right)}}{\epsilon}\sqrt{T}\right)$ | - | |
| Bastianello et al. [2024] | - | $\epsilon = \mathcal{O}\left(\frac{\zeta LV^2}{\beta^2\sigma^2}\left(1-e^{-\beta T}\right)\right)$ | convergent on $\beta$-strongly convex |
| **Ours** (Noisy-FedAvg) | $\sigma = o\left(\frac{V\sqrt{(\Phi^{-1}(\delta))^2+4\epsilon}}{\epsilon\sqrt{m}}\sqrt{2-\frac{1}{T}}\right)$ | $\epsilon = \mathcal{O}\left(\frac{\zeta V^2}{m\sigma^2}\left(2-\frac{1}{T}\right)\right)$ | convergent on non-convex |

## 5 Empirical Validation

**Setups.** We conduct experiments on MNIST [LeCun et al., 1998] and CIFAR-10 [Krizhevsky et al., 2009] with the LeNet-5 [LeCun et al., 1998] and ResNet-18 [He et al., 2016] models. We follow the widely used standard federated learning experimental setups to introduce heterogeneity by the Dirichlet splitting. The heterogeneity level is set high (Dir-0.1 splitting).

**Accuracy.** Table 4 shows the comparison on Noisy-FedAvg. Our theory precisely addresses this misconception and rigorously provides its privacy protection performance. It can be observed that as the number of clients increases, the impact of noise gradually diminishes. We have previously explained this principle: for the globally averaged model, the more noise involved in the averaging process, the closer it gets to the noise mean, which is akin to the situation without noise interference. When we adjust the intensity from $\sigma = 10^{-3}$ to $10^{-1}$, the accuracy decreases by $5.57\%$ and $1.62\%$ on $m = 20$ and $100$ respectively on the MNIST and $14.19\%$ and $11\%$ on the CIFAR-10. The local interval $K$ does not significantly affect noise, and the accuracy drops consistently. $K$ primarily affects global sensitivity and higher aggregation frequency usually means better performance.

**Sensitivity in Noisy-FedAvg.** We mainly study the impact from the scale $m$, local interval $K$, and clipping norm $V$, as shown in Fig. 2. The first figure clearly demonstrates the impact of the scale $m$ on sensitivity, which corresponds to the worst privacy bound $\mathcal{O}\left(\frac{1}{\sqrt{m}}\right)$. More clients generally imply stronger global privacy. The second figure shows evident that although increasing $K$ can raise the sensitivity during the process, it does not alter the upper bound of sensitivity after optimization converges. This is entirely consistent with our analysis, indicating that the privacy lower bound exists and is unaffected by $T$ and $K$. The third figure indicates that the sensitivity will be affected by the $V$, which corresponds to the worst privacy bound $\mathcal{O}(V)$.

Table 4: Comparison of the accuracy under different experimental settings. We select the scale $m$ from $[50, 100]$. Each client holds $600$ heterogeneous data samples of MNIST or $500$ heterogeneous data samples of CIFAR-10. For each scale, we test two settings of the local interval $K = 50, 100$, and $200$, respectively. Throughout the entire process, we fix $TK = 30000$. "-" means the training loss diverges. Each result is repeated $5$ times to compute its mean and variance.

| | Noisy Intensity | $m = 50$ | | | $m = 100$ | | |
|---|---|---|---|---|---|---|---|
| | | $K = 50$ | $K = 100$ | $K = 200$ | $K = 50$ | $K = 100$ | $K = 200$ |
| MNIST LeNet-5 | $\sigma = 1.0$ | - | - | - | - | - | - |
| | $\sigma = 10^{-1}$ | $95.40_{\pm 0.18}$ | $95.42_{\pm 0.15}$ | $95.21_{\pm 0.11}$ | $97.32_{\pm 0.14}$ | $97.50_{\pm 0.11}$ | $97.42_{\pm 0.18}$ |
| | $\sigma = 10^{-2}$ | $98.33_{\pm 0.12}$ | $98.02_{\pm 0.15}$ | $97.88_{\pm 0.12}$ | $98.71_{\pm 0.10}$ | $97.97_{\pm 0.08}$ | $97.72_{\pm 0.12}$ |
| | $\sigma = 10^{-3}$ | $98.41_{\pm 0.07}$ | $98.23_{\pm 0.03}$ | $98.00_{\pm 0.07}$ | $98.94_{\pm 0.04}$ | $98.50_{\pm 0.06}$ | $98.01_{\pm 0.10}$ |
| CIFAR-10 ResNet-18 | $\sigma = 1.0$ | - | - | - | - | - | - |
| | $\sigma = 10^{-1}$ | $53.76_{\pm 0.25}$ | $53.38_{\pm 0.23}$ | $53.49_{\pm 0.21}$ | $62.02_{\pm 0.28}$ | $61.33_{\pm 0.25}$ | $61.11_{\pm 0.17}$ |
| | $\sigma = 10^{-2}$ | $70.11_{\pm 0.22}$ | $69.08_{\pm 0.12}$ | $66.63_{\pm 0.16}$ | $74.34_{\pm 0.29}$ | $72.87_{\pm 0.19}$ | $70.74_{\pm 0.15}$ |
| | $\sigma = 10^{-3}$ | $70.98_{\pm 0.11}$ | $69.81_{\pm 0.20}$ | $67.98_{\pm 0.03}$ | $75.38_{\pm 0.19}$ | $74.44_{\pm 0.12}$ | $72.11_{\pm 0.06}$ |

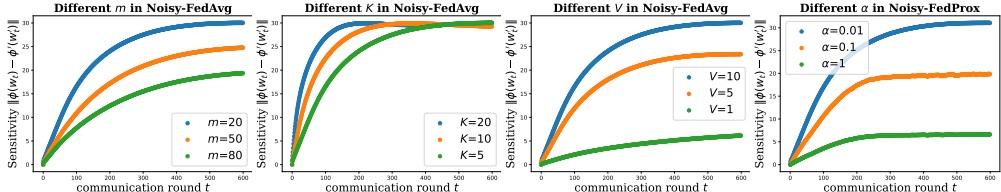

Figure 2: Sensitivity studies on `Noisy-FedAvg` and `Noisy-FedProx`. The general setups are $m = 20$, $K = 5$, and $V = 10$. In each group, we keep all other parameters fixed to ensure fairness.

**Sensitivity in `Noisy-FedProx`.** As shown in Fig. 2 (the fourth figure), the larger $\alpha$ means smaller global sensitivity. This is consistent with our analysis, which states that the lower bound of privacy performance is given by $\mathcal{O}\left(\frac{1}{\sqrt{\alpha}}\right)$. When we select $\alpha = 0$, it degrades to the `Noisy-FedAvg`

Table 5: Performance and sensitivity ($T = 600$).

| | Accuracy | Sensitivity |
|---|---|---|
| `Noisy-FedAvg` | 60.67 | 31.33 |
| `Noisy-FedProx` $\alpha = 0.01$ | 60.69 | 30.97 |
| `Noisy-FedProx` $\alpha = 0.1$ | **60.94** | **18.52** |
| `Noisy-FedProx` $\alpha = 1$ | 56.33 | 6.34 |

method. In fact, based on the comparison, we can see that when $\alpha$ is sufficiently small, i.e. $\alpha = 0.01$, its global sensitivity is almost at the same level as `Noisy-FedAvg`. In Table 5, we present a comparison between them. Although the proximal term provides limited improvement in accuracy, selecting an appropriate $\alpha$ significantly reduces global sensitivity. This implies that the privacy performance of `Noisy-FedProx` is far superior to that of `Noisy-FedAvg`. While achieving similar performance, the regularization proxy term can significantly reduce the global sensitivity of the output model, thereby enhancing privacy. This conclusion also demonstrates the superiority on privacy of a series of FL-DP optimization methods based on training with this regularization approach.

## 6 Summary

To our best knowledge, this paper is the first work to demonstrate convergent privacy for the general FL-DP paradigms. We comprehensively study and illustrate the fine-grained privacy level for `Noisy-FedAvg` and `Noisy-FedProx` methods based on $f$-DP analysis, an information-theoretic lossless DP definition. Moreover, we conduct comprehensive analysis with existing work on other DP frameworks and highlight the long-term cognitive bias of the privacy lower bound. Our analysis fills the theoretical gap in the convergent privacy of FL-DP while further providing a reliable theoretical guarantee for its privacy protection performance. Moreover, We conduct a series of experiments to verify the boundedness of global sensitivity and its influence on different variables, further validating that our theoretical analysis aligns more closely with practical scenarios.

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

**Limitations and Broader Impacts.** Our paper provides the first convergence-privacy analysis for the FL framework. The current analysis primarily includes the impact of multi-step updates on local nodes and the effect of multi-clients aggregation on the privacy bounds. A limitation of this paper is the inability to directly extend the privacy analysis to stability analysis. Stability analysis of convergence has always been a crucial theoretical objective in non-convex optimization. Although the trade-off function constructed by f-DP incorporates certain iterative properties of stability terms, it currently cannot directly derive convergence bounds for stability. Moreover, the theoretical analysis in this paper provides a crucial theoretical basis for privacy preservation, demonstrating that privacy can still be maintained under finite noise and infinitely long learning processes. This implies that many online methods can ensure privacy through cumulative noise accumulation, which may provide valuable guidance for privacy preservation in future engineering applications.

## A  General FL-DP Framework

FL framework usually allows local clients to train several iterations and then aggregates these optimized local models for global consistency guarantees. Though indirect access to the dataset significantly mitigates the risk of data leakage, vanilla gradients or parameters communicated to the server still bring privacy concerns, i.e. indirect leakage. Thus, DP techniques are introduced by adding isotropic noises on local parameters before communication, to further enhance privacy protection.

---

**Algorithm 1** General FL-DP Framework

---

**Input:** initial parameters $w_0$, round $T$, interval $K$
**Output:** global parameters $w_T$
1: **for** $t = 0, 1, 2, \cdots, T - 1$ **do**
2:    activate local clients and communications
3:    **for** client $i \in \mathcal{I}$ in parallel **do**
4:       set the initialization $w_{i,0,t} = w_t$
5:       **for** $k = 0, 1, 2, \cdots, K - 1$ **do**
6:          $w_{i,k+1,t} = \textit{L-update}(w_{i,k,t})$
7:       **end for**
8:       generate a noise $n_i \sim \mathcal{N}(0, \sigma^2 I_d)$
9:       communicate $w_{i,K,t} + n_i$ to the server
10:   **end for**
11:   $w_{t+1} = \textit{G-update}(\{w_{i,K,t} + n_i\})$
12: **end for**

---

In our analysis, we consider the FL-DP framework with the classical normal client-level noises, as shown in Algorithm 1. At the beginning of each communication round $t$, the server activates local clients and communicates necessary variables. Then local clients begin the training in parallel. We describe this process as a total of $K > 1$ steps of *L-update* function updates. Depending on algorithm designs, the specific form of local update functions varies. After training, the local clients enhance local privacy by adding noise perturbations to the uploaded model parameters. Our analysis primarily considers the properties of the isotropic Gaussian noise distribution, i.e. $n_i \sim \mathcal{N}(0, \sigma^2 I_d)$. Then the global server aggregates the noisy parameters to generate the global model $w_{t+1}$ via the *G-update* function. Repeat this for $T$ rounds and return $w_T$ as output.

## B  Preliminary Properties of $f$-DP

In this section, we mainly supplement some basic properties of $f$-DP, all of which are lemmas proposed by Dong et al. [2022]. Specifically, Lemmas 1 and 2 are employed in our theoretical analysis, whereas Lemmas 3 and 4 facilitate a lossless translation of our results into other standard DP frameworks for comparative purposes.

**Lemma 1 (Post-processing)** *If a randomized mechanism $\mathcal{M}$ is $f$-DP, any post processing mechanism based on $\mathcal{M}$ is still at least $f$-DP, i.e. $T(P'; Q') \geq T(P; Q)$ for any post-processing mapping which leads to $P \to P'$ and $Q \to Q'$.*

Intuitively, post-processing mappings bring some changes in the original distributions. However, such changes can not allow the updated distributions to be much easier to discern. This lemma also widely exists in other DP relaxations and stands as one of the foundational elements in current privacy analyses. In $f$-DP, this lemma also clearly demonstrates that the difficulty of hypothesis testing problems can not be simplified with the addition of known information, which still preserves the original distinguishability.

**Lemma 2 (Composition)** *We have a series of mechanisms $\mathcal{M}_i$ and a joint serial composition mechanism $\mathcal{M}$. Let each private mechanism $\mathcal{M}_i(\cdot, y_1, \cdots, y_{i-1})$ be $f_i$-DP for all $y_1 \in Y_1, \cdots, y_{i-1} \in Y_{i-1}$. Then the $n$-fold composed mechanism $\mathcal{M} : X \rightarrow Y_1 \times \cdots \times Y_n$ is $f_1 \otimes \cdots \otimes f_n$-DP, where $\otimes$ denotes the joint distribution. For instance, if $f = T(P; Q)$ and $g = T(P'; Q')$, then $f \otimes g = T(P \times P'; Q \times Q')$.*

The composition in the $f$-DP framework is *closed* and *tight*. This is also one of the advantages of privacy representation in $f$-DP. Correspondingly, the advanced composition theorem for $(\varepsilon, \delta)$-DP can not admit the optimal parameters to exactly capture the privacy in the composition process [Dwork et al., 2015]. However, the trade-off function has an exact probabilistic interpretation and can precisely measure the composition.

**Lemma 3 (GDP $\rightarrow$ ($\epsilon, \delta$)-DP)** *A $\mu$-GDP mechanism with a trade-off function $T_G(\mu)$ is also $(\epsilon, \delta(\epsilon))$-DP for all $\epsilon \geq 0$ where*

$$\delta(\epsilon) = \Phi\left(-\frac{\epsilon}{\mu} + \frac{\mu}{2}\right) - e^\epsilon \Phi\left(-\frac{\epsilon}{\mu} - \frac{\mu}{2}\right). \tag{20}$$

**Lemma 4 (GDP $\rightarrow$ RDP)** *A $\mu$-GDP mechanism with a trade-off function $T_G(\mu)$ is also $\left(\zeta, \frac{1}{2}\mu^2\zeta\right)$-RDP for any $\zeta > 1$.*

We state the transition and conversion calculations from $f$-DP (we specifically consider the GDP) to other DP relaxations, e.g. for the $(\varepsilon, \delta)$-DP and RDP. These lemmas can effectively compare our theoretical results with existing ones. Our comparison primarily aims to demonstrate that the convergent privacy obtained in our analysis would directly derive bounded privacy budgets in other DP relaxations. Moreover, we will illustrate how the convergent $f$-DP further addresses conclusions that current FL-DP work cannot cover theoretically, which provides solid support for understanding its reliability of privacy protection.

# C    Proof of Main Theorems

## C.1    Proofs of Theorem 1

We consider the general updates on the adjacent datasets $\mathcal{C}$ and $\mathcal{C}'$ on round $t$ as follows:

$$\begin{aligned}
w_{t+1} &= \phi(w_t) + \frac{1}{m}\sum_{i\in\mathcal{I}} n_{i,t}, \\
w'_{t+1} &= \phi'(w'_t) + \frac{1}{m}\sum_{i\in\mathcal{I}} n'_{i,t},
\end{aligned} \tag{21}$$

where $w_0$ is the initial state. $n_{i,t}$ and $n'_{i,t}$ are two noises generated from the normal distribution $\mathcal{N}(0, \sigma^2 I_d)$. To construct the interpolated sequence, we introduce the concentration coefficients $\lambda_t$ to provide a convex combination of the updates above, which is,

$$\widetilde{w}_{t+1} = \lambda_{t+1}\phi(w_t) + (1 - \lambda_{t+1})\phi'(\widetilde{w}_t) + \frac{1}{m}\sum_{i\in\mathcal{I}} n_{i,t}, \tag{22}$$

for $t = t_0, t_0+1, \cdots, T-1$. Furthermore, we set $\lambda_T = 1$ to let $\widetilde{w}_T = \phi(w_{T-1}) + \frac{1}{m}\sum_{i\in\mathcal{I}} n_{i,T-1} = w_T$, and we add the definition of $\widetilde{w}_{t_0} = w'_{t_0}$ as the interpolation beginning. $t_0$ determines the length of the interpolation sequence.

**Lemma 5** *According to the expansion of trade-off functions, for the general updates in Eq.(22), we have the following recurrence relation:*

$$T\left(\widetilde{w}_{t+1}; w'_{t+1}\right) \geq T\left(\widetilde{w}_t; w'_t\right) \otimes T_G\left(\frac{\sqrt{m}}{\sigma}\lambda_{t+1}\|\phi(w_t) - \phi'(\widetilde{w}_t)\|\right). \tag{23}$$

**Proof.** *Based on the post-processing and compositions, let $z$ and $z'$ be the corresponding noises above, for any constant $\lambda \in [0, 1]$, we have (subscripts are temporarily omitted):*

$$T\left(\lambda\phi(w) + (1-\lambda)\phi'(\widetilde{w}) + z; \phi'(w') + z'\right)$$
$$= T\left(\phi'(\widetilde{w}) + \lambda\left(\phi(w) - \phi'(\widetilde{w})\right) + z; \phi'(w') + z'\right)$$
$$\geq T\left(\left(\phi'(\widetilde{w}), \lambda\left(\phi(w) - \phi'(\widetilde{w})\right) + z\right); \left(\phi'(w'), z'\right)\right)$$
$$\geq T\left(\phi'(\widetilde{w}); \phi'(w')\right) \otimes T\left(\lambda\left(\phi(w) - \phi'(\widetilde{w})\right) + z; z'\right)$$
$$\geq T\left(\widetilde{w}; w'\right) \otimes T\left(\lambda\left(\phi(w) - \phi'(\widetilde{w})\right) + z; z'\right),$$

*where $z$ and $z'$ are two Gaussian noises that can be considered to be sampled from $\mathcal{N}(0, \frac{\sigma^2}{m}I_d)$ (average of $m$ isotropic Gaussian noises). Therefore, the distinguishability between the first term and the second term does not exceed the mean shift of the distribution, which is $\|\frac{\sqrt{m}}{\sigma}\lambda\left(\phi(w) - \phi'(\widetilde{w})\right)\|$. By taking $w = w_t$ and $\lambda = \lambda_{t+1}$, the proofs are completed.*

According to the above lemma, by expanding it from $t = t_0$ to $T-1$ and the factor $T(\widetilde{w}_{t_0}; w'_{t_0}) = T_G(0)$, we can prove the formulation in Eq. (10).

## C.2    Proofs of Theorem 2

Lemma 5 provides the general recursive relationship on the global states along the communication round $t$. To obtain the lower bound of the trade-off function, we only need to solve for the gaps $\|\phi(w) - \phi'(\widetilde{w})\|$. It is worth noting that the local update process here involves dual replacement of both the dataset ($\phi$ and $\phi'$) and the initial state ($w$ and $\widetilde{w}$). Therefore, we measure their maximum discrepancy by assessing their respective distances to the intermediate variable constructed by the cross-items:

$$\|\phi(w) - \phi'(\widetilde{w})\| \leq \underbrace{\|\phi(w) - \phi'(w)\|}_{\text{Data Sensitivity}} + \underbrace{\|\phi'(w) - \phi'(\widetilde{w})\|}_{\text{Model Sensitivity}}. \tag{24}$$

The first term measures the disparity in training on different datasets and the second term measures the gap in training from different initial models. One of our contributions is to provide their general gaps. In our paper, we expand the update function $\phi(x)$ by considering the multiple local iterations and federated cross-device settings. By simply setting the local interval to 1 and the number of clients to 1, our results can easily reproduce the original conclusion in [Bok et al., 2024]. Furthermore, our comprehensive considerations have led to a new understanding of the impact of local updates on privacy.

$\phi(w_t)$ and $\phi'(w_t)$ begin from $w_t$. $\phi'(w_t)$ and $\phi'(\widetilde{w}_t)$ adopt the data samples $\varepsilon' \in \mathcal{C}'$. We naturally use $w_{i,k,t}$ and $\widetilde{w}_{i,k,t}$ to represent individual states in $\phi(w_t)$ and $\phi'(\widetilde{w}_t)$, respectively. **To avoid ambiguity, we define the states in $\phi'(w_t)$ as $\hat{w}_{i,k,t}$.** When $i \neq i^\star$, since $\varepsilon = \varepsilon'$, then $w_{i,k,t}$ only differs from $\hat{w}_{i,k,t}$ on $i^\star$-th client.

**on the `Noisy-FedAvg` Method:**

**Lemma 6 (Data Sensitivity.)** *The **data sensitivity** caused by gradient descent steps can be bounded as:*

$$\|\phi(w_t) - \phi'(w_t)\| \leq \frac{2V}{m} \sum_{k=0}^{K-1} \eta_{k,t}, \tag{25}$$

*where $\eta_{k,t}$ is the learning rate at the $k$-th iteration of $t$-th communication round.*

**Proof.** *By directly expanding the update functions $\phi$ and $\phi'$ at $w_t$, we have:*

$$\|\phi(w_t) - \phi'(w_t)\|$$
$$= \|w_t - \frac{1}{m}\sum_{i\in\mathcal{I}}\sum_{k=0}^{K-1}\eta_{k,t}\nabla f_i(w_{i,k,t}, \varepsilon) - w_t + \frac{1}{m}\sum_{i\in\mathcal{I}}\sum_{k=0}^{K-1}\eta_{k,t}\nabla f_i(\hat{w}_{i,k,t}, \varepsilon')\|$$
$$\leq \frac{1}{m}\sum_{i\in\mathcal{I}}\sum_{k=0}^{K-1}\eta_{k,t}\|\nabla f_i(w_{i,k,t}, \varepsilon) - \nabla f_i(\hat{w}_{i,k,t}, \varepsilon')\|$$

$$= \frac{1}{m} \sum_{k=0}^{K-1} \eta_{k,t} \|\nabla f_{i^\star}(w_{i^\star,k,t}, \varepsilon) - \nabla f_{i^\star}(\hat{w}_{i^\star,k,t}, \varepsilon')\| \le \frac{2V}{m} \sum_{k=0}^{K-1} \eta_{k,t}.$$

*The last equation adopts $\varepsilon = \varepsilon'$ when $i \ne i^\star$. This completes the proofs.*

**Lemma 7 (Model Sensitivity.)** *The **model sensitivity** caused by gradient descent steps can be bounded as:*

$$\|\phi'(w_t) - \phi'(\widetilde{w}_t)\| \le (1 + \eta(K, t)L) \|w_t - \widetilde{w}_t\|, \tag{26}$$

*where $\eta(K,t) = \eta_{0,t} + \sum_{k=1}^{K-1} \eta_{k,t} \prod_{j=0}^{k-1} (1 + \eta_{j,t}L)$ is a constant related the selection of learning rates.*

**Proof.** *We first learn an individual case. On the $t$-th round, we assume the initial states of two sequences are $w_t$ and $\widetilde{w}_t$. Each is performed by the update function $\phi'$ for local $K$ steps. For each step, we have:*

$$\begin{aligned}
&\|\hat{w}_{i,k+1,t} - \widetilde{w}_{i,k+1,t}\| \\
&\le \|\hat{w}_{i,k,t} - \widetilde{w}_{i,k,t}\| + \eta_{k,t}\|\nabla f_i(\hat{w}_{i,k,t}, \varepsilon') - \nabla f_i(\widetilde{w}_{i,k,t}, \varepsilon')\| \\
&\le (1 + \eta_{k,t}L)\|\hat{w}_{i,k,t} - \widetilde{w}_{i,k,t}\|.
\end{aligned}$$

*This implies each gap when $k \ge 1$ can be upper bounded by:*

$$\|\hat{w}_{i,k,t} - \widetilde{w}_{i,k,t}\| \le (1 + \eta_{k-1,t}L)\|\hat{w}_{i,k-1,t} - \widetilde{w}_{i,k-1,t}\| \le \cdots \le \prod_{j=0}^{k-1}(1 + \eta_{j,t}L)\|w_t - \widetilde{w}_t\|.$$

*Then we consider the recursive formulation of the stability gaps along the iterations $k$. We can directly apply Eq.(22) to obtain the relationship for the differences updated from different initial states on the same dataset. By directly expanding the update function $\phi'$ at $w_t$ and $\widetilde{w}_t$, we have:*

$$\begin{aligned}
&\|\phi'(w_t) - \phi'(\widetilde{w}_t)\| \\
&= \|w_t - \frac{1}{m}\sum_{i\in\mathcal{I}}\sum_{k=0}^{K-1}\eta_{k,t}\nabla f_i(\hat{w}_{i,k,t}, \varepsilon') - \widetilde{w}_t + \frac{1}{m}\sum_{i\in\mathcal{I}}\sum_{k=0}^{K-1}\eta_{k,t}\nabla f_i(\widetilde{w}_{i,k,t}, \varepsilon')\| \\
&\le \|w_t - \widetilde{w}_t\| + \|\frac{1}{m}\sum_{i\in\mathcal{I}}\sum_{k=0}^{K-1}\eta_{k,t}\left(\nabla f_i(\hat{w}_{i,k,t}, \varepsilon') - \nabla f_i(\widetilde{w}_{i,k,t}, \varepsilon')\right)\| \\
&\le \|w_t - \widetilde{w}_t\| + \frac{L}{m}\sum_{i\in\mathcal{I}}\sum_{k=0}^{K-1}\eta_{k,t}\|\hat{w}_{i,k,t} - \widetilde{w}_{i,k,t}\| \\
&\le \left[1 + \left(\eta_{0,t} + \sum_{k=1}^{K-1}\eta_{k,t}\prod_{j=0}^{k-1}(1 + \eta_{j,t}L)\right)L\right]\|w_t - \widetilde{w}_t\|.
\end{aligned}$$

*This completes the proofs.*

We have successfully quantified the specific form of the problem as above. By solving for a series of reasonable values of the auxiliary variable $\lambda$ to minimize the above problem, we obtain the tight lower bound on privacy. Before that, let's discuss the learning rate to simplify this expression. Both $\eta(K,t)$ and $\sum \eta_{k,t}$ terms are highly related to the selections of learning rates. Typically, this choice is determined by the optimization process. Whether it's generalization or privacy analysis, both are based on the assumption that the optimization can converge properly. Therefore, we selected several different learning rate designs based on various combination methods to complete the subsequent analysis. Due to the unique two-stage learning perspective of federated learning, current methods for designing the learning rate generally choose between a constant rate or a rate that decreases with local rounds or iterations. Therefore, we discuss them separately including constant learning rate, cyclically decaying learning rate, stage-wise decaying learning rate, and continuously decaying learning rate. We provide a simple comparison in Figure 3.

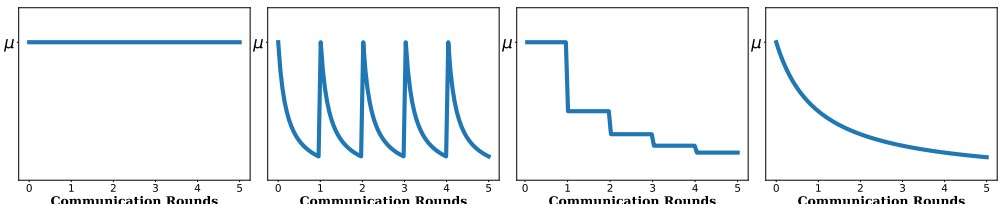

Figure 3: Four general setups of learning rate adopted in the federated learning community. From left to right, they are: *Constant learning rates*, *Cyclically decaying learning rates*, *Stage-wise decaying learning rate*, and *Continuously decaying learning rate*.

**Constant learning rates**  This is currently the simplest case. We consider the learning rate to always be a constant, i.e. $\eta_{k,t} = \mu$. Then we have that the accumulation term $\sum_{k=0}^{K-1} \eta_{k,t} = \mu K$. For the $\eta(K,t)$ term, we have:

$$\eta(K,t) = \eta_{0,t} + \sum_{k=1}^{K-1} \eta_{k,t} \prod_{j=0}^{k-1} (1 + \eta_{j,t}L) = \mu \sum_{k=0}^{K-1} (1 + \mu L)^k = \frac{1}{L}\left((1+\mu L)^K - 1\right).$$

When $K$ is selected, both of them can be considered as a constant related to $K$. The choice of $\mu$ also requires careful consideration. Although it is a constant, its selection is typically related to $m$, $K$, and $T$ based on the optimization process. We will discuss this point in the final theorems.

**Cyclically decaying learning rates**  Some works treat this learning process as an aggregation process of several local training processes, i.e. each local client learns from a better initial state (knowledge learned from other clients). And since the client pool is very large, most clients will exit after obtaining the model they desire. This setting is often used in "cross-device" scenarios [Kairouz et al., 2021]. Thus, local learning can be considered as an independent learning process. In this case, the learning rate is designed to decay in an inversely proportional function to achieve optimal local accuracy, i.e. $\eta_{k,t} = \frac{\mu}{k+1}$, and is restored to a larger initial value at the start of each round, i.e. $\eta_{0,t} = \mu$. Then we have the accumulation term:

$$\ln(K+1) = \int_{k=0}^{K} \frac{1}{k+1}dk \leq \sum_{k=0}^{K-1} \frac{1}{k+1} \leq 1 + \int_{0}^{K-1} \frac{1}{k+1}dk = 1 + \ln(K). \quad (27)$$

When $K$ is large, this term is dominated by $\mathcal{O}(\ln(K))$. Based on the fact that $K$ is very large in federated learning, we further approximate this term to $c\ln(K+1)$ where $c$ is a scaled constant. It is easy to check that there must exist $1 \leq c < 1.543$ for any $K \geq 1$. Thus we have the accumulation term as $\sum_{k=0}^{K-1} \eta_{k,t} = c\mu\ln(K+1)$. For the $\eta(K,t)$ term, we have its upper bound:

$$\eta(K,t) = \mu + \sum_{k=1}^{K-1} \frac{\mu}{k+1} \prod_{j=0}^{k-1} \left(1 + \frac{\mu L}{j+1}\right) \leq \mu + \sum_{k=1}^{K-1} \frac{\mu}{k+1} \prod_{j=0}^{k-1} \exp\left(\frac{\mu L}{j+1}\right)$$

$$= \mu + \sum_{k=1}^{K-1} \frac{\mu}{k+1} \left[\exp\left(\sum_{j=0}^{k-1} \frac{1}{j+1}\right)\right]^{\mu L} = \sum_{k=0}^{K-1} \frac{\mu}{k+1} \left[\exp\left(c\ln(k+1)\right)\right]^{\mu L}$$

$$= \mu \sum_{k=0}^{K-1} (k+1)^{c\mu L-1} \leq \mu \int_{k=0}^{K} (k+1)^{c\mu L-1} dk = \frac{1}{cL}\left((1+K)^{c\mu L} - 1\right).$$

The first inequality adopts $1 + x \leq e^x$ and the last adopts the concavity. Actually, we still can learn its general lower bound by a scaling constant. By adopting a scaling $b$, we can have $1 + x \geq e^{bx}$, which is equal to $b \leq \frac{\ln(x+1)}{x}$. It is also easy to check $0.693 < b < 1$ when $0 < x \leq 1$. Thus we have:

$$\eta(K,t) = \mu + \sum_{k=1}^{K-1} \frac{\mu}{k+1} \prod_{j=0}^{k-1} \left(1 + \frac{\mu L}{j+1}\right) \geq \mu + \sum_{k=1}^{K-1} \frac{\mu}{k+1} \prod_{j=0}^{k-1} \exp\left(\frac{\mu bL}{j+1}\right)$$

$$= \mu + \sum_{k=1}^{K-1} \frac{\mu}{k+1} \left[ \exp\left( \sum_{j=0}^{k-1} \frac{1}{j+1} \right) \right]^{\mu bL} = \sum_{k=0}^{K-1} \frac{\mu}{k+1} \left[ \exp\left( c\ln(k+1) \right) \right]^{\mu bL}$$

$$= \mu \sum_{k=0}^{K-1} (k+1)^{c\mu bL-1} \geq \mu \int_{k=-1}^{K-1} (k+1)^{c\mu bL-1}\, dk = \frac{1}{cbL} K^{c\mu bL}.$$

The last inequality also adopts concavity. Through this simple scaling, we learn the general bounds for the learning rate function $\eta(K, t)$ as:

$$\frac{1}{cbL} K^{c\mu bL} \leq \eta(K, t) \leq \frac{1}{cL} \left( (1+K)^{c\mu L} - 1 \right), \tag{28}$$

where $1 \leq c < 1.543$, $0.693 < b < 1$ and $\mu \leq \frac{1}{L}$ (this condition is almost universally satisfied in current optimization theories). Although we cannot precisely find the tight bound of this function $\eta(K, t)$, we can still treat it as a form based on constants to complete the subsequent analysis, i.e. it could be approximated as a larger upper bound $\frac{1}{L} \left( (1+K)^{c\mu L} - 1 \right)$. More importantly, we have determined that this learning rate function still diverges as $K$ increases.

**Stage-wise decaying learning rates** This is one of the most common selections of learning rate in the current federated community, which is commonly applied in "cross-silo" scenarios [Kairouz et al., 2021]. When the client pool is not very large, clients who participate in the training often aim to establish long-term cooperation to continuously improve their models. Therefore, each client will contribute to the entire training process over a long period. From a learning perspective, local training is more like exploring the path to a local optimum rather than actually achieving the local optimum. Therefore, each local training will adopt a constant learning rate and perform several update steps, i.e. $\eta_{k,t} = \eta_t$. At each communication round, the learning rate decays once and continues to the next stage, i.e. $\eta_t = \frac{\mu}{t+1}$. Based on the analysis of the constant learning rate, the accumulation term is $\sum_{k=0}^{K-1} \eta_{k,t} = \frac{\mu K}{t+1}$. For the $\eta(K, t)$ term, we have:

$$\eta(K, t) = \frac{\mu}{t+1} + \sum_{k=1}^{K-1} \frac{\mu}{t+1} \prod_{j=0}^{k-1} \left( 1 + \frac{\mu L}{t+1} \right)$$

$$= \frac{\mu L}{t+1} \sum_{k=0}^{K-1} \left( 1 + \frac{\mu L}{t+1} \right)^{k} = \frac{1}{L} \left( \left( 1 + \frac{\mu L}{t+1} \right)^{K} - 1 \right).$$

It can be seen that the analysis of this function is more challenging because the learning rate function $\eta(K, t)$ is decided by $t$, which introduces complexity to the subsequent analysis. We will explain this in detail in the subsequent discussion.

**Continuously decaying learning rates** This is a common selection of learning rate in the federated community, involving dual learning rate decay along both local training and global training. This can almost be applied to all methods to adapt to the final training, including both the cross-silo and cross-device cases. At the same time, its analysis is also more challenging because the learning rate is coupled with communication rounds and local iterations, yielding new upper and lower bounds. We consider the general case $\eta_{k,t} = \frac{\mu}{tK+k+1}$. Therefore, the accumulation term can be bounded as:

$$\sum_{k=0}^{K-1} \frac{1}{tK+k+1} > \int_{k=0}^{K} \frac{1}{tK+k+1} dk = \ln\left( \frac{tK+K+1}{tK+1} \right),$$

$$\sum_{k=0}^{K-1} \frac{1}{tK+k+1} < \frac{1}{tK+1} + \int_{k=0}^{K-1} \frac{1}{tK+k+1} dk = \frac{1}{tK+1} + \ln\left( \frac{tK+K}{tK+1} \right).$$

Similarly, when $K$ is large enough, this term is dominated by $\mathcal{O}\left( \ln\left( \frac{t+1}{t} \right) \right)$. For simplicity in the subsequent proof, we follow the process above and let it be $z\ln\left( \frac{t+2}{t+1} \right)$ to include the term at $t=0$. It is also easy to check that $z > 1$ is a constant for any $K > 1$. And $z$ is also a constant. It means

we can always select the lower bound as its representation. Therefore, for the learning rate function $\eta(K, t)$, we have:

$$\eta(K, t) = \frac{\mu}{tK + 1} + \sum_{k=1}^{K-1} \frac{\mu}{tK + k + 1} \prod_{j=0}^{k-1} \left(1 + \frac{\mu L}{tK + j + 1}\right)$$

$$\leq \frac{\mu}{tK + 1} + \sum_{k=1}^{K-1} \frac{\mu}{tK + k + 1} \left[\exp\left(\sum_{j=0}^{k-1} \frac{1}{tK + j + 1}\right)\right]^{\mu L}$$

$$= \frac{\mu}{tK + 1} + \sum_{k=1}^{K-1} \frac{\mu}{tK + k + 1} \left[\exp\left(z \ln\left(\frac{tK + k + 1}{tK + 1}\right)\right)\right]^{\mu L}$$

$$= \frac{\mu}{(tK + 1)^{z\mu L}} \sum_{k=0}^{K-1} (tK + k + 1)^{z\mu L - 1}$$

$$\leq \frac{\mu}{(tK + 1)^{z\mu L}} \int_{k=0}^{K} (tK + k + 1)^{z\mu L - 1} \, dk = \frac{1}{zL} \left(\left(\frac{tK + K + 1}{tK + 1}\right)^{z\mu L} - 1\right).$$

Similarly, we introduce the coefficient $b$ to provide the lower bound as:

$$\eta(K, t)$$

$$= \frac{\mu}{tK + 1} + \sum_{k=1}^{K-1} \frac{\mu}{tK + k + 1} \prod_{j=0}^{k-1} \left(1 + \frac{\mu L}{tK + j + 1}\right)$$

$$\geq \frac{\mu}{tK + 1} + \sum_{k=1}^{K-1} \frac{\mu}{tK + k + 1} \left[\exp\left(\sum_{j=0}^{k-1} \frac{1}{tK + j + 1}\right)\right]^{\mu b L}$$

$$= \frac{\mu}{tK + 1} + \sum_{k=1}^{K-1} \frac{\mu}{tK + k + 1} \left[\exp\left(z \ln\left(\frac{tK + k + 1}{tK + 1}\right)\right)\right]^{\mu b L}$$

$$= \frac{\mu}{(tK + 1)^{z\mu b L}} \sum_{k=0}^{K-1} (tK + k + 1)^{z\mu b L - 1}$$

$$\geq \frac{\mu}{(tK + 1)^{z\mu b L}} \int_{k=-1}^{K-1} (tK + k + 1)^{z\mu b L - 1} \, dk = \frac{1}{zbL} \left(\left(\frac{tK + K}{tK + 1}\right)^{z\mu b L} - \left(\frac{tK}{tK + 1}\right)^{z\mu b L}\right)$$

$$> \frac{1}{zbL} \left(\left(\frac{tK + K}{tK + 1}\right)^{z\mu b L} - 1\right).$$

Through the sample scaling, we learn the general bounds for the learning rate function $\eta(K, t)$ as:

$$\frac{1}{zbL} \left(\left(\frac{tK + K}{tK + 1}\right)^{z\mu b L} - 1\right) < \eta(K, t) \leq \frac{1}{zL} \left(\left(\frac{tK + K + 1}{tK + 1}\right)^{z\mu L} - 1\right), \qquad (29)$$

where $1 < z$, $0.693 < b < 1$ and $\mu \leq \frac{1}{L}$. Obviously, when $K$ is large enough, the learning rate term is still dominated by $\mathcal{O}\left(\left(\frac{t+2}{t+1}\right)^{z\mu L} - 1\right)$. Therefore, to learn the general cases, we can consider the specific form of the learning rate function based on the constant scaling as $\frac{1}{L}\left(\left(\frac{t+2}{t+1}\right)^{z\mu L} - 1\right)$. As $t$ increases, this function will approach zero.

**on the `Noisy-FedProx` Method:**

In this part, we will address the differential privacy analysis of a noisy version of another classical federated learning optimization method, i.e. the `Noisy-FedProx` method. The vanilla `FedProx` method is an optimization algorithm designed for cross-silo federated learning, particularly to address the challenges caused by data heterogeneity across different clients. Unlike traditional federated learning algorithms like `FedAvg`, which can struggle with variations in data distribution, it introduces a proximal term to the objective function. This helps to stabilize the training process and improve convergence. Specifically, it adopts the consistency as the penalized term to correct the local objective:

$$\min_w f_i(w) + \frac{\alpha}{2}\|w - w_t\|^2. \tag{30}$$

The proximal term is a very common regularization term in federated learning and has been widely used in both federated learning and personalized federated learning approaches. It introduces an additional penalty to the local objective, ensuring that local updates are optimized towards the local optimal solution while being subject to an extra global constraint, i.e. each local update does not stray too far from the initialization point. In fact, there are many optimization methods that apply such regularization terms. For example, various federated primal-dual methods based on the `ADMM` approach construct local Lagrangian functions, and in personalized federated learning, local privatization regularization terms are introduced to differentiate from the vanilla consistency objective. The analysis of the above methods is fundamentally based on a correct understanding of the advantages and significance of the proximal term in stability error. In this paper, to achieve a cross-comparison while maintaining generality, we consider the optimization process of local training as total $K$-step updates:

$$\phi(w_t) = w_t - \frac{1}{m}\sum_{i\in\mathcal{I}}\sum_{k=0}^{K-1}\eta_{k,t}\left(\nabla f_i(w_{i,k,t},\varepsilon) + \alpha\left(w_{i,k,t} - w_t\right)\right). \tag{31}$$

Here, we also employ the proofs mentioned in the previous section, and our study of the difference term is based on both data sensitivity and model sensitivity perspectives. We provide these two main lemmas as follows.

**Lemma 8 (Data Sensitivity.)** *The local data sensitivity of the `Noisy-FedProx` method at $t$-th communication round can be upper bounded as:*

$$\|\phi(w_t) - \phi'(w_t)\| \leq \frac{2V}{m\alpha}. \tag{32}$$

**Proof.** *We first consider a single step in Eq.(31) as:*

$$w_{i,k+1,t} = w_{i,k,t} - \eta_{k,t}\left(\nabla f_i(w_{i,k,t},\varepsilon) + \alpha(w_{i,k,t} - w_t)\right).$$

*The proximal term brings more opportunities to enhance the analysis of local updates. We can split the proximal term and subtract the $w_t$ term on both sides, resulting in a recursive formula for the cumulative update term:*

$$w_{i,k+1,t} - w_t = (1 - \eta_{k,t}\alpha)\left(w_{i,k,t} - w_t\right) - \eta_{k,t}\nabla f_i(w_{i,k,t},\varepsilon).$$

*The above equation indicates that a reduction factor $1 - \eta_{k,t}\alpha < 1$ can limit the scale of local updates. This is a very good property, allowing us to shift the analysis of the data sensitivity to their relationship of local updates. According to the above, we can upper bound the gaps between $\{w_{i,k,t}\}$ and $\{\hat{w}_{i,k,t}\}$ sequences as:*

$$\begin{aligned}
&\|(w_{i,k+1,t} - w_t) - (\hat{w}_{i,k+1,t} - w_t)\| \\
&= \|\left(1 - \eta_{k,t}\alpha\right)\left[(w_{i,k,t} - w_t) - (\hat{w}_{i,k,t} - w_t)\right] - \eta_{k,t}(\nabla f_i(w_{i,k,t},\varepsilon) - \nabla f_i(\hat{w}_{i,k,t},\varepsilon'))\| \\
&\leq (1 - \eta_{k,t}\alpha)\|(w_{i,k,t} - w_t) - (\hat{w}_{i,k,t} - w_t)\| + \eta_{k,t}\|\nabla f_i(w_{i,k,t},\varepsilon) - \nabla f_i(\hat{w}_{i,k,t},\varepsilon')\| \\
&\leq (1 - \eta_{k,t}\alpha)\|(w_{i,k,t} - w_t) - (\hat{w}_{i,k,t} - w_t)\| + 2\eta_{k,t}V.
\end{aligned}$$

*Different from proofs in Lemma 6, the term $1 - \eta_{k,t}\alpha$ can further decrease the stability gap during accumulation. By summing form $k = 0$ to $K - 1$, we can obtain:*

$$\|(w_{i,K,t} - w_t) - (\hat{w}_{i,K,t} - w_t)\|$$

$$\leq \prod_{k=0}^{K-1} (1 - \eta_{k,t}\alpha) \, \|(w_{i,0,t} - w_t) - (\hat{w}_{i,0,t} - w_t)\| + \sum_{k=0}^{K-1} \left( \prod_{j=k+1}^{K-1} (1 - \eta_{j,t}\alpha) \right) 2\eta_{k,t}V$$

$$= 2V \sum_{k=0}^{K-1} \left( \prod_{j=k+1}^{K-1} (1 - \eta_{j,t}\alpha) \right) \eta_{k,t}.$$

Here, we provide a simple proof using a constant learning rate to demonstrate that its upper bound can be independent of $K$. By considering $\eta_{k,t} = \mu$, we have:

$$\sum_{k=0}^{K-1} \left( \prod_{j=k+1}^{K-1} (1 - \eta_{j,t}\alpha) \right) \eta_{k,t} = \sum_{k=0}^{K-1} \left( \prod_{j=k+1}^{K-1} (1 - \mu\alpha) \right) \mu = \frac{1 - (1 - \mu\alpha)^K}{\alpha} < \frac{1}{\alpha}.$$

In fact, when the learning rate decays with $k$, it can still be easily proven to have a constant upper bound. Therefore, in the subsequent proofs, we directly use the form of this constant upper bound as the result of data sensitivity in the `Noisy-FedProx` method. Based on the definition of $\phi(w)$, we have:

$$\|\phi(w_t) - \phi'(w_t)\| = \| \left( \phi(w_t) - w_t \right) - \left( \phi'(w_t) - w_t \right) \| = \|\frac{1}{m} \sum_{i \in \mathcal{I}} [(w_{i,K,t} - w_t) - (\hat{w}_{i,K,t} - w_t)] \|$$

$$= \frac{1}{m} \| \left( w_{i^\star,K,t} - w_t \right) - \left( \hat{w}_{i^\star,K,t} - w_t \right) \| < \frac{2V}{m\alpha}.$$

This completes the proofs.

**Lemma 9 (Model Sensitivity.)** *The local model sensitivity of the `Noisy-FedProx` method at $t$-th communication round can be upper bounded as:*

$$\|\phi'(w_t) - \phi'(\widetilde{w}_t)\| \leq \frac{\alpha}{\alpha_L} \|w_t - \widetilde{w}_t\|. \tag{33}$$

**Proof.** *We also adopt the splitting above. Since both sequences are trained on the same dataset, the gradient difference can be measured by the parameter difference. Therefore, we directly consider the form of the parameter difference:*

$$\|\hat{w}_{i,k+1,t} - \widetilde{w}_{i,k+1,t}\|$$
$$= \|(1 - \eta_{k,t}\alpha)(\hat{w}_{i,k,t} - \widetilde{w}_{i,k,t}) - \eta_{k,t}(\nabla f_i(\hat{w}_{i,k,t}, \varepsilon') - \nabla f_i(\widetilde{w}_{i,k,t}, \varepsilon')) - \eta_{k,t}\alpha(w_t - \widetilde{w}_t)\|$$
$$\leq (1 - \eta_{k,t}\alpha)\|\hat{w}_{i,k,t} - \widetilde{w}_{i,k,t}\| + \eta_{k,t}L\|\hat{w}_{i,k,t} - \widetilde{w}_{i,k,t}\| + \eta_{k,t}\alpha\|w_t - \widetilde{w}_t\|$$
$$= (1 - \eta_{k,t}\alpha_L)\|\hat{w}_{i,k,t} - \widetilde{w}_{i,k,t}\| + \eta_{k,t}\alpha\|w_t - \widetilde{w}_t\|,$$

*where $\alpha_L = \alpha - L$ is a constant. Here, we consider $\alpha > L$. When $\alpha \leq L$, its upper bound can not be guaranteed to be reduced. When $\alpha > L$, it can restore the property of decayed stability. By summing from $k = 0$ to $K - 1$, we can obtain:*

$$\|\hat{w}_{i,K,t} - \widetilde{w}_{i,K,t}\|$$
$$\leq \prod_{k=0}^{K-1} (1 - \eta_{k,t}\alpha_L)\|\hat{w}_{i,0,t} - \widetilde{w}_{i,0,t}\| + \sum_{k=0}^{K-1} \left( \prod_{j=k+1}^{K-1} (1 - \eta_{k,t}\alpha_L) \right) \eta_{k,t}\alpha\|w_t - \widetilde{w}_t\|$$
$$= \left[ \prod_{k=0}^{K-1} (1 - \eta_{k,t}\alpha_L) + \sum_{k=0}^{K-1} \left( \prod_{j=k+1}^{K-1} (1 - \eta_{k,t}\alpha_L) \right) \eta_{k,t}\alpha \right] \|w_t - \widetilde{w}_t\|.$$

*Similarly, we learn the upper bound from a simple constant learning rate. By select $\eta_{k,t} = \mu$, we have:*

$$\prod_{k=0}^{K-1} (1 - \eta_{k,t}\alpha_L) + \sum_{k=0}^{K-1} \left( \prod_{j=k+1}^{K-1} (1 - \eta_{k,t}\alpha_L) \right) \eta_{k,t}\alpha$$

$$= \prod_{k=0}^{K-1}(1-\mu\alpha_L) + \sum_{k=0}^{K-1}\left(\prod_{j=k+1}^{K-1}(1-\mu\alpha_L)\right)\mu\alpha$$

$$= (1-\mu\alpha_L)^K + \alpha\frac{1-(1-\mu\alpha_L)^K}{\alpha_L}$$

$$= \frac{\alpha}{\alpha_L} - \frac{L(1-\mu\alpha_L)^K}{\alpha_L} < \frac{\alpha}{\alpha_L}.$$

*The same, it can also be checked that the general upper bound of the stability gaps is a constant even if the learning rate is selected to be decayed along iteration $k$. Therefore, in the subsequent proofs, we directly use the form of this constant upper bound as the result of model sensitivity in the* `Noisy-FedProx` *method. Based on the definition of $\phi(w)$, we have:*

$$\|\phi'(w_t) - \phi'(\widetilde{w}_t)\| = \|\frac{1}{m}\sum_{i\in\mathcal{I}}(\hat{w}_{i,K,t} - \widetilde{w}_{i,K,t})\| \le \frac{1}{m}\sum_{i\in\mathcal{I}}\|\hat{w}_{i,K,t} - \widetilde{w}_{i,K,t}\| \le \frac{\alpha}{\alpha_L}\|w_t - \widetilde{w}_t\|.$$

*This completes the proofs.*

### C.3 Solution of Eq. (13)

According to the recurrence relation in Lemma 5, we can confine the privacy amplification process to a finite number of steps with the aid of an interpolation sequence, yielding to the convergent bound. Therefore, we have:

$$T(w_T; w_T') = T(\widetilde{w}_T; w_T')$$

$$\ge T(\widetilde{w}_{T-1}; w_{T-1}') \otimes T_G\left(\frac{\sqrt{m}}{\sigma}\lambda_T\|\phi(w_{T-1}) - \phi'(\widetilde{w}_{T-1})\|\right)$$

$$\ge T(\widetilde{w}_{t_0}; w_{t_0}') \otimes \cdots \otimes T_G\left(\frac{\sqrt{m}}{\sigma}\lambda_T\|\phi(w_{T-1}) - \phi'(\widetilde{w}_{T-1})\|\right)$$

$$= T(w_{t_0}'; w_{t_0}') \otimes T_G\left(\frac{\sqrt{m}}{\sigma}\sqrt{\sum_{t=t_0}^{T-1}\lambda_{t+1}^2\|\phi(w_t) - \phi'(\widetilde{w}_t)\|^2}\right)$$

$$\ge T_G\left(\frac{\sqrt{m}}{\sigma}\sqrt{\sum_{t=t_0}^{T-1}\lambda_{t+1}^2(\rho_t\|w_t - \widetilde{w}_t\| + \gamma_t)^2}\right).$$

Although the above form appears promising, an inappropriate selection of the key parameters will still cause divergence due to the recurrence term coefficient $1 + \eta(K,t)L > 1$, leading it to approach infinity as $t$ increases. For instance, small $t_0$ will result in a significantly increased $\lambda$ and the bound will be closed to the stability gap $\|w_T - w_T'\|$, and large $t_0$ will result in a long accumulation of the stability gaps, which is also unsatisfied. At the same time, it is also crucial to choose appropriate $\lambda$ to ensure that the stability accumulation can be reasonably diluted. Therefore, we also need to thoroughly investigate how significant the stability gap caused by the interpolation points is. According to Eq.(21) and (22), we have:

$$\|w_{t+1} - \widetilde{w}_{t+1}\| \le (1-\lambda_{t+1})(\rho_t\|w_t - \widetilde{w}_t\| + \gamma_t).$$

The above relationship further constrains the stability of the interpolation sequence. It is worth noting that the upper bound of the final step is independent of the choice of $\lambda$. At the same time, since all terms are positive, given a group of specific $\lambda$, taking the upper bound at each possible $t$ will result in the maximum error accumulation. This is also the worst-case privacy we have constructed. Therefore, solving the worst privacy could be considered as solving the following problem:

$$\underbrace{\min_{\{\lambda_{t+1}\},t_0}\underbrace{\max_{\{\|w_t-\widetilde{w}_t\|\}}\sum_{t=t_0}^{T-1}\lambda_{t+1}^2(\rho_t\|w_t - \widetilde{w}_t\| + \gamma_t)^2}_{\text{worst privacy}}}_{\text{tight privacy lower bound}}, \tag{34}$$

$$\text{s.t. } \|w_{t+1} - \widetilde{w}_{t+1}\| \le (1-\lambda_{t+1})(\rho_t\|w_t - \widetilde{w}_t\| + \gamma_t).$$

Based on the above analysis, this problem can be directly transformed into a privacy minimization problem when the interpolation sequence reaches the maximum stability error. Therefore, we just need to solve the following problem:

$$\min_{\{\lambda_{t+1}\},t_0} \sum_{t=t_0}^{T-1} \lambda_{t+1}^2 \left( \gamma_t \|w_t - \widetilde{w}_t\| + \gamma_t \right)^2, \tag{35}$$

$$\text{s.t. } \|w_{t+1} - \widetilde{w}_{t+1}\| = (1 - \lambda_{t+1}) \left( \rho_t \|w_t - \widetilde{w}_t\| + \gamma_t \right).$$

It is important to note that this upper bound condition is usually loose because the probability that the interpolation terms simultaneously reach their maximum deviation is very low. This is merely the theoretical worst-case privacy scenario.

Then we solve the minimization problem. By considering the worst stability conditions, we can provide the relationship between the gaps and coefficients $\lambda_{t+1}$ as:

$$\|w_{t+1} - \widetilde{w}_{t+1}\| = \rho_t \|w_t - \widetilde{w}_t\| + \gamma_t - \lambda_{t+1} \left( \rho_t \|w_t - \widetilde{w}_t\| + \gamma_t \right).$$

Expanding it from $t = t_0$ to $T$, we have:

$$0 = \|w_T - \widetilde{w}_T\| = \left( \prod_{t=t_0}^{T-1} \rho_t \right) \|w_{t_0} - \widetilde{w}_{t_0}\| + \sum_{t=t_0}^{T-1} \left( \prod_{j=t+1}^{T-1} \rho_j \right) \left[ \gamma_t - \lambda_{t+1} \left( \rho_t \|w_t - \widetilde{w}_t\| + \gamma_t \right) \right].$$

Due to the term $\lambda_{t+1} \left( \rho_t \|w_t - \widetilde{w}_t\| + \gamma_t \right)$ being part of the analytical form of the minimization objective, we preserve the integrity of this algebraic form and only split it from the perspectives of coefficients $\lambda_t$, $\rho_t$ and $\gamma_t$. According to the definition $\widetilde{w}_{t_0} = w'_{t_0}$, then we have:

$$\sum_{t=t_0}^{T-1} \left( \prod_{j=t+1}^{T-1} \rho_j \right) \lambda_{t+1} \left( \rho_t \|w_t - \widetilde{w}_t\| + \gamma_t \right) = \left( \prod_{t=t_0}^{T-1} \rho_t \right) \|w_{t_0} - w'_{t_0}\| + \sum_{t=t_0}^{T-1} \left( \prod_{j=t+1}^{T-1} \rho_j \right) \gamma_t. \tag{36}$$

The above equation presents the summation of the term $\lambda_{t+1} \left( \rho_t \|w_t - \widetilde{w}_t\| + \gamma_t \right)$ accompanied by a scaling coefficient $\left( \prod_{j=t+1}^{T-1} \rho_j \right) > 1$. It naturally transforms the summation form into an initial stability gap and a constant term achieved through a combination of learning rates. To solve it, we can directly adopt the Cauchy-Schwarz inequality to separate the terms and construct a constant term based on the form of the scaling coefficient to find its achievable lower bound:

$$\sum_{t=t_0}^{T-1} \lambda_{t+1}^2 \left( \rho_t \|w_t - \widetilde{w}_t\| + \gamma_t \right)^2$$

$$\geq \left( \sum_{t=t_0}^{T-1} \left( \prod_{j=t+1}^{T-1} \rho_j \right) \lambda_{t+1} \left( \rho_t \|w_t - \widetilde{w}_t\| + \gamma_t \right) \right)^2 \left( \sum_{t=t_0}^{T-1} \left( \prod_{j=t+1}^{T-1} \rho_j \right)^2 \right)^{-1}$$

$$= \left( \left( \prod_{t=t_0}^{T-1} \rho_t \right) \|w_{t_0} - w'_{t_0}\| + \sum_{t=t_0}^{T-1} \left( \prod_{j=t+1}^{T-1} \rho_j \right) \gamma_t \right)^2 \left( \sum_{t=t_0}^{T-1} \left( \prod_{j=t+1}^{T-1} \rho_j \right)^2 \right)^{-1}.$$

Although the original problem requires solving the $\lambda_{t+1}$, here we can know one possible minimum form of the problem no longer includes this parameter. In fact, this parameter has been transformed into the optimality condition of the Cauchy-Schwarz inequality.

Therefore, we only need to optimize it w.r.t the parameter $t_0$. Unfortunately, this part highly correlates with the stability gaps $\|w_{t_0} - w'_{t_0}\|$. Current research progress indicates that in non-convex optimization, this term diverges as the number of training rounds $t$ increases. This makes it difficult for us to accurately quantify its specific impact on the privacy bound. If $t_0$ is very small, it means that the introduced stability gap will also be very small. However, consequently, the coefficients of the $\rho_t$ and $\gamma_t$ terms will increase due to the accumulation over $T - t_0$ rounds. To detail this, we have to make certain compromises. Because $t_0$ is an integer belonging to $[0, T-1]$, we denote its optimal selection by $t^\star$ (it certainly exists when $T$ is given). Therefore, the privacy lower bound under other

choices of $t_0$ will certainly be more relaxed, i.e. $\text{Privacy}_{t_0} \leq \text{Privacy}_{t^\star}$ (privacy is weak at other selection of $t_0$). This allows us to look for other asymptotic solutions instead of finding the optimal solution. Although we cannot ultimately achieve the form of the optimal solution, we can still provide a stable privacy lower bound. To eliminate the impact of stability error, we directly choose $t_0 = 0$, yielding the following bound:

$$
\begin{aligned}
\mathcal{H}_\star \leq \mathcal{H}_0 &= \left( \left( \prod_{t=t_0}^{T-1} \rho_t \right) \| w_{t_0} - w'_{t_0} \| + \sum_{t=t_0}^{T-1} \left( \prod_{j=t+1}^{T-1} \rho_j \right) \gamma_t \right)^2 \left( \sum_{t=t_0}^{T-1} \left( \prod_{j=t+1}^{T-1} \rho_j \right)^2 \right)^{-1} \Bigg|_{t_0=0} \\
&= \left( \sum_{t=0}^{T-1} \left( \prod_{j=t+1}^{T-1} \rho_j \right) \gamma_t \right)^2 \left( \sum_{t=0}^{T-1} \left( \prod_{j=t+1}^{T-1} \rho_j \right)^2 \right)^{-1}.
\end{aligned}
$$

By substituting the values of $\rho_t$ and $\gamma_t$ under different cases, then we can prove the main theorems in this paper.

