# OpenReview forum: "Convergent Differential Privacy Analysis for General Federated Learning"
_NeurIPS.cc/2025/Conference — Submitted to NeurIPS 2025_

### Official Review · Reviewer_4A4P · 2025-06-15

**Clarity:** 3
**Significance:** 3
**Originality:** 3
**Rating:** 5
**Confidence:** 3

**Summary:**

To narrow the gap between the theoretical privacy guarantee and the empirical privacy guarantee of FL-DP framework, this paper established sharper privacy bounds for two federated learning algorithms i.e., noise-FedAvg and noise-FedProx, under non-convex and smooth losses based on the $f$-DP. Inspired by the shifted interpolation technique used in the f-DP analysis for centralized noise gradient descent and its variants, this paper extends this technique to federated learning and proves that the noise-FedAvg and noise-FedProx enjoy a tight convergent lower bound and a constant lower bound, respectively. The theoretical analyses and results are also verified on a series of experiments.

**Questions:**

There are many federated learning algorithms that have been proved to converge faster than FedAvg, such as SCAFFOLD [1], FedAdam, FedAdagrad [2]. FedProx is a personalized federated learning algorithm, thereby being significantly different from standard federated learning algorithms, such as FedAvg, SCAFFOLD. Why did this paper select noise-FedAvg and noise-FedProx as examples, rather than noise-SCAFFOLD or noise-FedAdam? Can the current privacy analysis be extended to the other noised federated learning algorithms? In my opinion, it would be better to prove that there exists a noise-SCAFFOLD, or a noise-FedAdam that attains a strong privacy guarantee.

References

[1]Karimireddy et al. SCAFFOLD: Stochastic Controlled Averaging for Federated Learning. ICML, 2020.

[2]Reddi et al. Adaptive Federated Optimization. ICLR, 2021.

**Ethical Concerns:**

["NO or VERY MINOR ethics concerns only"]

**Final Justification:**

This paper established sharper privacy bounds for two FL-DP algorithms, resolving an important open problem on the privacy guarantee of FL-DP framework. I believe this makes significant contributions to the federated learning community. The authors' rebuttal has addressed my two questions: first, why this paper selects noise-FedAvg and noise-FedProx as examples, rather than noise-SCAFFOLD or noise-FedAdam; and second, whether the current privacy analysis can be extended to the other noised federated learning algorithms. Thus, I decide to raise my score.

**Limitations:**

yes

**Paper Formatting Concerns:**

I do not notice any major formatting issues in this paper.

**Quality:**

3

**Strengths And Weaknesses:**

Strengths:

This paper established sharper privacy bounds for two FL-DP algorithms, resolving an important open problem on the privacy guarantee of FL-DP framework. I believe this makes significant contributions to the federated learning community.

Weaknesses.

There is a lack of analysis on the convergence rate to a stationary point for the noise-FedAvg and noise-FedProx algorithms. It is known that there is a trade-off between the privacy and utility in differential privacy algorithms. Although this paper established sharper privacy bounds on the noise-FedAvg and noise-FedProx algorithms, it is unclear what the convergence rates to a stationary point are for the two algorithms. For example, in Theorem 3, this paper gives various privacy lower bound by tuning the learning rates of noise-FedAvg. However, there is a lack of convergence rates under various learning rates.

---

> ### Author Rebuttal · Authors · 2025-07-31
>
> We greatly appreciate the positive feedback and for raising a insightful academic question. We have provided a detailed response to this issue below. We have broken down your question into the following sub-questions and will address them one by one.
>
> ## A1: FL vs PFL.
>
> We thank the reviewer for bringing up this point. While there may be room for academic discussion regarding the categorization of FL methods, we would like to clarify the classification context adopted in our work during the writing process.
>
> **FL vs PFL**
>
> FL generally focuses on training a global model by:
> $$
> \min_{w} \frac{1}{m}\sum_{i}f_i(w),
> $$
> and PFL generally focuses on training a list of local model by:
> $$
> \min_{w_0,w_1,\cdots,w_t} \frac{1}{m}\sum_{i}f_i(x_i) + \phi(w_0, w_1, \cdots, w_m),
> $$
> where $\phi(\cdot)$ is a reguralization term. Usually, classical terms like $\sum_{i,j} \Vert w_i - w_j \Vert^2$ or $\sum_i \Vert w_i - \overline{w}_i \Vert^2$ are adopted to address the consistency.
>
> **What does FedProx do?**
>
> FedProx solves the FL optimization problem instead of PFL's. It transfers the general FL problem to a consistency-constrained optimization problem:
> $$
> \min_{w} \frac{1}{m}\sum_{i}f_i(w_i), \quad s.t. \quad w_i=w.
> $$
> Subsequently, this method employs a penalty-based approach (adopting $\Vert w_i - w \Vert^2$ as the penalization of equation $w_i = w$) to **alternately** optimize the local surrogate function and the consistency constraint:
> $$
> \min_{w_i} f_i(w_i) + \frac{\beta}{2}\Vert w_i - w \Vert^2 \quad \text{and} \quad \min_{w} \frac{1}{m}\sum_{i}\Vert w_i - w \Vert^2
> $$
>
> We agree that FedProx shares similarities with certain personalized PFL methods. However, **in our work, we classify FL based on whether the optimization objective is a single global model $w$, which is a strict optimization definition**. From this perspective, FedProx remains a standard FL method, as its goal is still to optimize a global model across clients. This definition is also consistent with the original description in the their paper [P1].
>
> [P1] Federated Optimization in Heterogeneous Networks
>
> ## Why did this paper select noise-FedAvg and noise-FedProx as examples?
>
> We thank the reviewer for raising this question. Our core objective is to address the challenge that interpolation techniques developed for SGD cannot be directly applied to FL, due to the presence of multi-step local training on local clients.To overcome this, we introduce a new technique that performs shifted interpolation at the global level, and proposes a finer-grained sensitivity analysis at the local level by decoupling model and data sensitivity. This enables us to capture the privacy dynamics of FL training more precisely under practical conditions.
>
> We select the FedAvg and FedProx because they represent two main branches from the perspective of FL optimization:
>
> (a) directly solve the primal problem (like **FedAvg**, SCAFFOLD, FedAdam)
>
> (b) solve the surrogate problem or the primal-dual problem (like **FedProx**, FedPD, FedDyn, FedADMM)
>
> These two classes of methods are fundamentally disjoint in their analytical frameworks, as they adopt different assumptions and requirements when handling local training procedures. Consequently, their corresponding optimization analyses also differ significantly. The goal of this work is to provide the first convergent privacy analysis for two mainstream optimization paradigms in FL, thereby offering strong theoretical support for understanding and improving its privacy guarantees.
>
> We believe the reviewer's suggestion is highly valuable. A more comprehensive exploration of variance reduction techniques and globally adaptive methods used in FL could indeed deepen our understanding of the evolving privacy capabilities of FL. However, as the first work to provide convergent privacy guarantees for FL, we would like to focus on two foundational algorithms that represent the mainstream optimization paradigms in the field. **In our view, if privacy cannot be guaranteed even in these fundamental settings, it would be even more challenging to ensure it in their advanced variants.** This is also the main reason why we chose to start from fundamental algorithmic research.
>
> **We truly believe that engaging in further discussion with you will help us improve our submission. We are very happy to share more insights and clarifications regarding our work, and would be glad to continue the discussion should you have any further questions.**

---

> > ### Comment · Reviewer_4A4P · 2025-08-03
> >
> > I thank the authors for their thorough rebuttal, which has addressed my questions. I have no more questions.
> >
> > I will decide whether to raise my score after discussing with other reviewers.

---

> > > ### Author Response · Authors · 2025-08-03
> > >
> > > We are glad that the discussion helps clarify the concerns. In the final version, we will include a definition of the objective and explicitly state the scope of our theoretical analysis. Once again, we sincerely thank you for the time and effort you dedicated to reviewing our submission, as well as for your valuable suggestions.

---

### Official Review · Reviewer_ByT8 · 2025-06-28

**Clarity:** 2
**Significance:** 3
**Originality:** 2
**Rating:** 4
**Confidence:** 3

**Summary:**

The paper provides a theoretical analysis for differential privacy in federated learning. Based on f-DP analysis, the paper evaluates the worst privacy of DP-FedAvg and DP-FedProx. Specifically, the paper adopts the shifted interpolation sequence to enable a privacy amplification analysis and analyze the global sensitivity. Compared with other studies, the paper is the first to provide a convergent DP analysis for non-convex functions in DP federated learning.

**Questions:**

1. Could the proposed analysis be extended to the client sampling setting?
2. What are the main challenges when adopting f-DP and shifted interpolation technique to analyze DP-FL?
3. Is there any new insight for analyzing DP-FL?

**Ethical Concerns:**

["NO or VERY MINOR ethics concerns only"]

**Final Justification:**

I have read the authors' rebuttal. I suggest that the analysis on the client-sampling case could be added in the final version, which is a usual case in federated learning. I'll keep my positive score.

**Limitations:**

The analysis assumes the full-participation setting in federated learning. However, client sampling is usually adopted.

**Paper Formatting Concerns:**

N.A.

**Quality:**

3

**Strengths And Weaknesses:**

Strengths:
1. The paper provides the first convergent DP analysis in DP-FL methods with non-convex functions.
2. The analysis includes different kinds of learning rate decaying policies for DP-FedAvg.
3. The results can be converted to other DP analytical frameworks.


Weaknesses:
1. The analysis assumes the full-participation setting, while client sampling is usually adopted in federated learning.
2. The paper does not introduce any new techniques for DP analysis.
3. The challenges of analyzing DP-FL with f-DP is unclear.

---

> ### Author Rebuttal · Authors · 2025-07-31
>
> We sincerely thank the reviewers for their positive feedback and valuable suggestions. In this round of discussion, we will address each of the raised questions in detail.
>
> ## Q1: Could the proposed analysis be extended to the client sampling setting?
>
> We sincerely thank the reviewer for raising this important question. The core objective of our work is to demonstrate that FL methods can maintain provable privacy guarantees even under long local training and multi-client settings. More importantly, we show that with carefully chosen hyperparameters, FL can naturally provide stronger privacy protection than standard centralized SGD, aligning with the original design philosophy of FL.
>
> It is worth emphasizing that privacy guarantees are even stronger under partial client participation, because when the client containing the neighboring datapoint is not selected in a round, the global sensitivity is solely determined by the amplified model sensitivity, without any contribution from local data sensitivity.
>
> Therefore, it suffices to apply a simple probability-weighted correction to the model sensitivity lemma in order to extend our results to the partial participation setting. This extension is incremental and natural, and we will explicitly include a remark after each main theorem to discuss the implications under partial client participation.
>
>
> ## Q2: What are the main challenges when adopting f-DP and shifted interpolation technique to analyze DP-FL?
>
> We appreciate the insightful comment on this matter. We believe that the main challenge lies in the fact that interpolation-based analyses developed for Noisy-SGD cannot be directly applied to local training in FL. Such analysis must simultaneously account for both the number of communication rounds $T$ and the local update interval $K$. In contrast to standard Noisy-SGD, where interpolation can be applied per step, local training in FL requires using a fixed interpolation coefficient at each aggregation step to ensure model correctness. This constraint significantly complicates the process of identifying the worst-case privacy. In fact, our early attempts also indicate that a direct extension of interpolation analysis under this setting is nearly intractable.
>
> To bridge the gap caused by the heterogeneous nature of multi-node local training in FL, we have further developed auxiliary technique that explicitly measure data and model sensitivities during local training (Figure 1 (right) and Section 4.2 of our paper). Additionally, due to the difficulty of solving the minimax problem for the worst-case privacy loss, we propose a relaxation approach (Section 4.3). Together, these contributions the convergent privacy theory to the FL.
>
>
> ## Q3: Is there any new insight for analyzing DP-FL?
>
> We thank the reviewer for raising this exploratory question. To the best of our knowledge, our work provides the first convergent privacy analysis under the non-convex FL training paradigm. Furthermore, our results provide two insightful points to help dispel two common misconceptions about privacy in FL:
>
> (a) **Privacy loss does can still converge with the number of local invertals**, contrary to the common belief that longer local training necessarily leads to weaker privacy.
>
> (b) **FL naturally provides stronger privacy guarantees than standard SGD**, since SGD corresponds to the special case where the number of $m=1$ and $K=1$. In fact, as the total number of clients increases, the privacy amplification becomes weaker due to the reduced per-client influence.
>
> These two findings clarify that the privacy advantages of FL are not merely due to the absence of direct access to local data, but are instead supported by rigorous theoretical guarantees. Our analysis thus reinforces the original motivation behind FL design, providing a principled foundation for its privacy-preserving properties as it continues to evolve. In addition, we provide a theoretical justification for the privacy benefits of a commonly used proxy regularization term in FL, which further deepens the understanding of privacy dynamics during local model training.
>
> **We truly believe that engaging in further discussion with you will help us improve our submission. We are very happy to share more insights and clarifications regarding our work, and would be glad to continue the discussion should you have any further questions.**

---

> ### Comment · Reviewer_ByT8 · 2025-08-04
>
> Thank you for your response. My concerns have been addressed. I suggest that the authors add the discussion about the client sampling setting in the final version. I'll keep my positive rating.

---

> > ### Author Response · Authors · 2025-08-04
> >
> > We sincerely appreciate your positive comments and valuable suggestions on our work. In the next version, we will include additional discussion on the sampling strategy and its connection with existing theories, as well as provide a more comprehensive comparison with DP-SGD. Thank you again for your constructive feedback and for dedicating your time to review our paper.

---

### Official Review · Reviewer_6nCw · 2025-07-02

**Clarity:** 3
**Significance:** 2
**Originality:** 2
**Rating:** 4
**Confidence:** 4

**Summary:**

The paper tackles a long-standing open question in DPFL: whether a constant noise variance can still guarantee privacy when training runs for arbitrarily many communication rounds.
Using the shifted interpolation technique, the authors derive tight, convergent privacy lower bounds for two canonical algorithms:
- Noisy-FedAvg: under four learning-rate schedules
- Noisy-FedProx: showing the proximal term $\alpha$ yields privacy that converges to a constant independent of the local-update interval $K$.
They also provide empirical studies on MNIST and CIFAR-10 that verify the bounded global sensitivity predicted by the theory.

**Questions:**

- How big is the loss when you force $t_{0}=0$ in the $\lambda_{t}$ optimisation? Does this gap fade as the ratio $T/t_{0}$ gets large?
- If each client picks its own $\alpha$, does the nice independent of $K$ privacy result still hold?
- Can some comments be given about possibilities in incorporating adaptive per-round clipping into current sensitivity analysis?
- Can more validation regarding $m$ be provided (more different values of $m$). See also the Weakness part.
- Can some comments address what would happen if $\alpha<L$ and possibilities to avoid the condition? See also the first point of Weakness.


Minor points:
- Line 263, a typo for $m=20$?

**Ethical Concerns:**

["NO or VERY MINOR ethics concerns only"]

**Final Justification:**

After reading the responses to reviews, I decide to raise my score.

**Limitations:**

This is a theoretical work with no obvious limitation.

**Paper Formatting Concerns:**

None.

**Quality:**

3

**Strengths And Weaknesses:**

Strengths & Contributions


- First stable privacy bound for non-convex FL
- Splitting global sensitivity into data and model terms yields intuitive, tunable bounds
- Experiments on MNIST/CIFAR-10 verify bounded global sensitivity and the effect of parameters.


Limitations & Weaknesses

- The condition $\alpha>L$ (Theorem 4) seems a little strong. Can some comments address what would happen otherwise? Also, in the experiments, the optimality is taken at, for instance in Table 5, $\alpha = 0.1$, which is a quite small value and seems impossible for $L$.
- All runs use fewer than 100 clients and vision data. Moreover, the authors claim a theoretical relationship between privacy and client number $m$, but only reported $m\in \{50, 100\}$ in experiments. More validation would be more convincing.
- The analysis relies largely on an application of the shifted interpolation technique on the federated setting.

---

> ### Author Rebuttal · Authors · 2025-07-31
>
> We sincerely appreciate the reviewer's conscientious efforts and the insightful academic suggestions. We will provide detailed responses to each of the issues raised in the rebuttal discussion.
>
> ## Q1: How big is the loss when you force $t_0=0$ in the $\lambda_t$ optimization? Does this gap fade as the ratio $T/t_0$ gets large?
>
> We sincerely appreciate the reviewer for raising this highly professional academic question. The validity of our relaxation can be supported by comparison with existing studies, particularly the discussion in Bok et al., 2024 regarding DP-SGD. The core difficulty lies in the fact that, when measuring the worst-case privacy loss, the minimization of the privacy w.r.t. both $t_0$ and ${\lambda_t}$ is a NP-hard problem. To achieve effective estimation, Bok et al. directly assume that the diameter of the entire optimization space is $D$. Therefore, the global sensitivity at any $t_0$ is upper bounded by $D$, which is a constant.
>
> In contrast, our proof avoids making such an assumption. We directly evaluate the outcome of the relaxed formulation. Regarding the reviewer's concern about the tightness of our bound, we would like to clarify that when $m=1$ and $K=1$, FL-DP degrades to DP-SGD, and our results is consistent with the bound presented in Bok et al when $T/t_0\rightarrow\infty$.
>
> **Why not assume that the global sensitivity at $t_0$ is a constant?**
>
> While this is an optimization trick, this projection is seldom adopted in real-world training. If $t_0$ were an identifiable constant, then this assumption would certainly hold. However, if $t_0$ grows with the total number of steps $T$, the assumption may no longer be valid. Although our proof introduces a relaxation in the upper bound, comparison with their results shows that the complexity of our bound differs from theirs by only a constant factor. As $T/t_0\rightarrow\infty$, this gap vanishes and becomes small enough.
>
> ## Q2: If each client picks its own $\alpha$, does the nice independent of $K$ privacy result still hold?
>
> We sincerely thank the reviewer for raising this question. When each private client is associated with a different $\alpha_i$, we need to consider two cases. The first case is straightforward, namely, $\min(\alpha_0, \alpha_1, \cdots,\alpha_m) > L$, then, by defining the notation $\alpha=\min(\alpha_0, \alpha_1, \cdots,\alpha_m)$, the results/theorems still hold. A more refined expression for $\alpha_i$ can also be obtained, but this does not affect the main conclusion (it only changes the result by a constant factor).
>
> The other case is relatively more complex, namely when there exists $\alpha_i < L$. We note that the reviewer also mentioned the analysis of $\alpha < L$ in Q5, and the discussion of these two cases overlaps. Therefore, we defer the analysis to our response to Q5.
>
> ## Q3: Can some comments be given about possibilities in incorporating adaptive per-round clipping into current sensitivity analysis?
>
> We thank the reviewer for the great question! We first provide an intuitive understanding of the gradient clipping parameter $V$. According to its definition, gradient clipping works as $g_{\text{clip}}=\frac{g}{max(1, \Vert g\Vert/V)}$, which makes:
>
> (case 1) if $\Vert g\Vert \leq V$, $g = g$;
>
> (case 2) if $\Vert g\Vert > V, g = V\cdot g/\Vert g\Vert$.
>
> One of its practical roles is to ensure stability during training by preventing sudden large gradients from causing training to crash. Another important role is to bound the maximum update distance per round, thereby providing a stable upper bound for the change in global sensitivity. In current practice, the vast majority of gradient clipping methods use a constant threshold.
>
> In existing theoretical analyses, it is common to use Case 2 as a unified upper bound that covers both Case 1 and Case 2, which implies that the global sensitivity of any single-step update is upper bounded by $O(\eta_t V)$. Therefore, any adaptive change to $V$ can be equivalently represented through a change in the learning rate $\eta_t$.
>
> If, in the future, more refined assumptions on the gradients can be made, such that Case 2 is no longer used as a unified upper bound when measuring global sensitivity, and instead Case 1 is further decomposed with greater precision, then incorporating adaptive $V$ would become highly meaningful.
>
> The above shares our current understanding about this question. As this is still an open problem, we would greatly appreciate any further insights or suggestions from the reviewer in the discussion.
>
> ## Q4: Can more validation regarding be provided (more different values of $m$). See also the Weakness part.
>
> We sincerely thank the reviewer for raising this question. We have supplemented our experiments with a smaller setting of $m=20$ and a larger setting of $m=500$, and further evaluated the behavior of global sensitivity under $m=500$. Due to the recent change in the NeurIPS rebuttal policy, we presented the results using multi-point sampling in a tabular format instead of figures, which may be less intuitive. We would be happy to further discuss the experimental results if the reviewer has any questions or concerns.
>
> **m=20 and 500**
>
> We evaluate the different $K$ and $\sigma$ when $m=20$ and $500$.
>
> |     | $m=20, K=50$ | $m=20, K=100$ | $m=20, K=200$ | $m=500, K=50$ | $m=500, K=100$ | $m=500, K=200$ |
> |:---:|:-----:|:-----:|:-----:|:-----:|:-----:|:-----:|
> |$\sigma=1.0$ | - | - | - | - | - | - |
> |$\sigma=0.1$ | 42.69 (0.33) | 41.69 (0.28) | 41.94 (0.35) | 68.42 (0.49) | 66.93 (0.41) | 66.36 (0.55) |
> |$\sigma=0.01$| 58.99 (0.18) | 58.59 (0.14) | 55.62 (0.27) | 77.68 (0.13) | 76.25 (0.19) | 73.84 (0.23) |
> |$\sigma=0.001$| 60.23 (0.08) | 59.35 (0.13) | 56.13 (0.18) | 78.14 (0.22) | 77.32 (0.34) | 76.07 (0.17) |
>
> We increase the number of clients to 500 to measure how the sensitivity changes with respect to time $T$.
>
> |     | T=100 | T=200 | T=300 | T=400 | T=500 | T=600 |
> |:---:|:-----:|:-----:|:-----:|:-----:|:-----:|:-----:|
> |m=20 |  16.73|  24.55|  27.14|  28.69|  29.63|  30.02|
> |m=50 |  10.62|  16.94|  20.45|  23.13|  23.88|  24.04|
> |m=500|  4.47 |  6.98 |  7.75 |  8.42 |  8.67 |  8.77 |
>
> It can be observed that the convergence trend aligns with our analysis.
>
> **More Results on ViT-Small**
>
> We conducted tests on ViT-Small, and the results show a similar trend to ResNet-18, although the sensitivity is higher, the convergence behavior remains largely consistent.
>
> setup: m=50, K=5, V=10, to ensure satisfactory convergence of the ViT-Small model, we increased the number of training steps to $T=1000$
>
> |     | T=200 | T=400 | T=600 | T=800 | T=1000 |
> |:---:|:-----:|:-----:|:-----:|:-----:|:-----:|
> |ResNet-18 |  16.94|  23.13| 24.04|  24.11|  24.17|
> |ViT-Small|  23.52 |  31.37 |  35.66 |  36.32 |  36.58 |
>
> Sensitivity of ResNet-18 converges after approximately 600 steps, while that of ViT-Small converges after 800 steps.
>
> ## Q5: Can some comments address what would happen if $\alpha < L$ and possibilities to avoid the condition? See also the first point of Weakness.
>
> We thank the reviewer for raising this question. $\alpha > L$ is an ideal condition that enables the Noisy-FedProx method to maintain a convergent worst-case privacy guarantee even under a **constant learning rate**. When $\alpha < L$, we can ensure the convergence of the privacy upper bound by using an **iteratively decaying learning rate**, similar to the ID case of Noisy-FedAvg in Table 1.
>
> Our lemma 8 is not affected. We provide the proof of Lemma 9 under a decaying learning rate. For the line 758, Since $\alpha < L$ is negative in this case. For the clarity, we define $\alpha_s = -\alpha_L > 0$. Therefore, we have:
> $$
> \eta(K,t) = \prod_{k=0}^{K-1}(1+\eta_{k,t}\alpha_s)+\sum_{k=0}^{K-1}(\prod_{j=k+1}^{K-1}(1+\eta_{k,t}\alpha_s))\eta_{k,t}\alpha.
> $$
> Let $\eta_{k,t}=\frac{\mu}{tK+k+1}$ be the iteratively decaying learning rate, the upper bound of the second term can be derived by referring to the corresponding proof under the same learning rate in Noisy-FedAvg, where this term has been bounded independently of $K$. Here, we provide the proof of the the first term:
> $$
> \begin{align*}
> \prod_{k=0}^{K-1}(1+\eta_{k,t}\alpha_s) \leq \exp(\alpha_s\mu\sum_{k=0}^{K-1}\frac{1}{tK+k+1}) < \exp(\alpha_s\mu\int_{-1}^{K-1}\frac{1}{tK+k+1}dk) = \exp(\alpha_s\mu (\ln(tK+K)-\ln(tK))) = (\frac{t+1}{t})^{\mu\alpha_s}.
> \end{align*}
> $$
> Similarly, this term remains independent of $K$. Substituting the full expression of $\eta(K,t)$, we can easily show that the Noisy-FedProx method achieves a convergent privacy that is slightly smaller but of comparable complexity to that of the Noisy-FedAvg method. Of course, the proofs for both methods are essentially very similar in nature.
>
> **We summarize the conclusions as follows for Noisy-FedProx.**
>
> |     | Independent on $K$? | condition required |
> |:---:|:-----:|:-----:|
> | $\alpha > L$ | yes | decreasing or constant learning rate |
> | $\alpha \leq L$| yes | iteratively decaying $\eta_{k,t}=\frac{\mu}{tK+k+1}$ |
>
> **Then we answer the case of adaptive $\alpha_i<L$ in Q2.**
>
> We recommend using iteratively decaying learning rate for training, which can also ensure convergent privacy guarantees.
>
> We would like to emphasize that the condition $\alpha > L$ is an additional property we identified in our analysis, which ensures privacy convergence even under relaxed learning rate schedules. **However, this does not imply that privacy convergence strictly depends on this condition.** With a decaying learning rate, privacy convergence can always be guaranteed.
>
> ## Typos
>
> We will correct this in the next version; the value should be $m = 50$. Additionally, we will include experiments for $m = 20$ and $m = 500$.
>
> **We truly believe that engaging in further discussion with you will help us improve our submission. We are very happy to share more insights and clarifications regarding our work, and would be glad to continue the discussion should you have any further questions.**

---

> > ### Comment · Reviewer_6nCw · 2025-08-06
> >
> > I thank the authors for their thorough rebuttal, which has addressed most of my concerns. However, I will decide whether to raise my score after discussing with other reviewers.

---

> > > ### Author Response · Authors · 2025-08-06
> > >
> > > We are glad to have addressed the reviewer’s concerns. The issues you raised regarding the adaptive proxy coefficient and the scenario where $\alpha < L$ are of great academic significance. We will update the final version with two new remarks specifically discussing these aspects. If you have any further questions, we would be more than happy to share our understanding. Once again, we sincerely thank you for your time and effort in reviewing our work.

---

### Official Review · Reviewer_eb6Y · 2025-07-02

**Clarity:** 4
**Significance:** 3
**Originality:** 2
**Rating:** 4
**Confidence:** 4

**Summary:**

This paper addresses a limitation in the analysis of privacy guarantees in Federated Learning with Differential Privacy (FL-DP). It challenges the common belief that privacy bounds necessarily diverge over time. By applying f-DP and a shifted interpolation technique, the authors analyze the privacy of Noisy-FedAvg and Noisy-FedProx under non-convex settings. The authors theoretically demonstrate that these methods can achieve convergent privacy bounds — a claim that prior composition-based analyses fail to provide. The authors support their theory with empirical evaluations of sensitivity and accuracy under different configurations.

**Questions:**

Q1 - How do the contributions of this paper go beyond prior analyses of DP-SGD, and what is truly novel in the FL context? The paper leverages the f-DP framework and shifted interpolation technique introduced by others. Works like Ye & Shokri (2022) and Altschuler & Talwar (2022) already showed convergent privacy for (strongly) convex losses, and a recent result by Chien & Li (2025) proved a convergent RDP bound for non-convex DP-SGD without smoothness assumptions. The authors should clarify exactly what new challenges are solved here that were not in those works. For example, does the multi-client, multi-step setting require new lemmas or insights beyond a straightforward combination of known results? Is the FedProx analysis (with a constant lower bound on ε) entirely novel? A detailed discussion of how this work differs from and improves upon concurrent works (especially in handling client heterogeneity or proving Theorem 4 for FedProx) would strengthen the paper’s originality.

Q2 - In Section 4.3, the analysis introduces an optimization problem to bound the trade-off function and then makes a relaxation (using $H_0$ in place of the intractable $H^*$) to obtain a solvable form. Could the authors clarify the impact of this relaxation on the final privacy bound? Is the resulting bound provably tight (i.e., matches the true worst-case privacy loss) or is there a gap? The paper claims the relaxed solution $H_0$ still achieves a convergent constant privacy loss, but it’s not clear if this is merely an existence proof or a tight characterization. For example, if one were to compute the exact privacy loss via another method, would it match the expressions in Table 1? Understanding this is important for trusting the results – if the relaxation introduces only negligible looseness, that’s fine, but if it’s significantly loose, the “tight convergent bound” claim might need nuance. A clear explanation (perhaps with theoretical or empirical evidence of tightness) would increase my confidence in the results.

Q3 –The experimental results are encouraging, but I have suggestions to bolster them. (a) Can the authors provide a comparison to classical privacy accounting in the experiments? For instance, take one of the FL setups and compute the $(\epsilon,\delta)$ after T rounds using the Moments Accountant or RDP composition, and compare it to the $\epsilon$ predicted by their f-DP analysis (converted to $(\epsilon,\delta)$). This would empirically demonstrate the improvement in the privacy bound (e.g., showing that naive composition might give $\epsilon > 10$ while the new method gives a bounded $\epsilon \approx 2$ for large $T$, hypothetically). (b) Consider evaluating on a larger scale or different domain, if possible. The current experiments use at most 100 clients and relatively small models; would the privacy convergence hold in, say, a language modeling task with hundreds of clients, or an image classification with a deeper network? While I understand resource and space constraints, even a discussion of this (or simulated results) would be valuable.

Q4 –The analysis reveals interesting trade-offs, particularly with the proximal coefficient α in FedProx. The experiments (Table 5 and Fig.2d) show that larger α drastically reduces sensitivity (hence improves privacy) but can hurt accuracy if α is too large. Could the authors comment on how to choose α optimally in practice? Is there a recommended range where we get most privacy benefit with minimal accuracy drop (e.g., α≈0.1 in their CIFAR-10 experiment improved privacy a lot with negligible accuracy loss)? This guidance would help practitioners balance the utility-privacy trade-off using FedProx. Similarly, for Noisy-FedAvg, do the results suggest an optimal clipping norm $V$ or number of local steps $K$ to use from a privacy perspective (since $V$ and $K$ affect the bounds)? Any intuition or recommendations here would enhance the practical relevance of the theory.

**Ethical Concerns:**

["NO or VERY MINOR ethics concerns only"]

**Limitations:**

Yes

**Quality:**

3

**Strengths And Weaknesses:**

Strength – Quality & Rigor: The paper is technically solid and presents a rigorous theoretical analysis. It builds on state-of-the-art privacy tools — the f-DP framework and shifted interpolation method — which together provide a tight characterization of privacy loss. The proofs appear correct and are grounded in recent advances (e.g. using the exact composition in f-DP instead of loose $(\epsilon,\delta)$ composition). The authors carefully handle the complexities of federated settings (like multi-step local updates and heterogeneous data) that previously foiled direct analysis. The theoretical results are clearly stated (with theorems for each scenario and a helpful summary table) and are novel in that they cover non-convex objectives and realistic FL settings. The quality of writing and organization is good – the paper motivates the problem clearly in the introduction.

Strength – Significance: This work addresses a previously unresolved question: Can federated learning with noise maintain privacy guarantees indefinitely? The findings may be meaningful to the FL and DP communities. By refuting the “divergent $\epsilon$” issue, the paper instills confidence that deploying DP in federated training is viable for many rounds – bridging a gap between theory and the encouraging empirical evidence. It also provides practical insights: for instance, the result that adding a proximal term (FedProx) yields strictly better privacy at little cost is valuable for practitioners designing private FL algorithms.

Strength – Clarity: The paper is generally well-written and structured. The authors articulate the privacy problem and their solution approach in a way that readers (familiar with DP) can follow. Mathematical notations (like trade-off functions $T(P;Q)$) are defined, and important assumptions (L-smoothness, clipping norm, noise variance) are stated clearly. The inclusion of multiple learning rate scenarios and discussion of each in the theorems improves clarity, as it shows how different training strategies affect privacy.

Weakness – Originality: The main concern is that the methodological novelty is limited. The core analytical tools employed — f-DP (trade-off function approach) and the shifted interpolation technique — are not introduced by this paper but by prior works (Dong et al., 2022; Bok et al., 2024). Essentially, the authors apply these existing techniques to the federated learning setting, instead of developing a fundamentally new privacy analysis method. While doing so is non-trivial (due to new challenges like client heterogeneity and local steps) and the paper is the first to handle the FL case, one could view the contribution as an incremental extension of earlier analyses of DP-SGD. Indeed, recent research has already shown convergent privacy loss for centralized DP-SGD under various conditions: e.g., Altschuler & Talwar (2022) and Ye & Shokri (2022) proved convergent bounds for convex problems, and very recently Chien & Li (ICLR 2025) extended this to non-convex, non-smooth losses in the single-machine (non-federated) scenario (https://openreview.net/pdf?id=kjmLabjSE2#:~:text=with%20respect%20to%20the%20number,Our%20analysis%20relies%20on). Compared to those, this work focuses on the federated multi-client scenario. Thus, the novelty lies in tailoring known techniques to FL and examining FedProx’s effect, rather than inventing new theory from scratch. This is a modest advance in originality.

Weakness – Empirical Evaluation: While the experiments support the theoretical claims, their scope is somewhat limited. The authors validate the key insights on vision datasets (MNIST, CIFAR-10) with relatively simple models, which is a good start, but the evaluation could be more comprehensive. For instance, all experiments fix a total of $T \times K = 30000$ iterations; this ensures a baseline for comparison, but it would be interesting to see privacy/utility trade-offs for truly long training (very large $T$) explicitly, or on more complex tasks (e.g., NLP or large-scale federated benchmarks). Additionally, the paper does not report actual $(\epsilon,\delta)$ values or compare against classical composition bounds in the experiments. It would strengthen the work to empirically demonstrate how much tighter the new analysis is – e.g., by showing that for a given $\delta$, the effective $\epsilon$ under the new analysis is significantly lower than that predicted by a moments accountant or RDP composition after the same training duration. The current experiments focus on accuracy and sensitivity metrics, which are proxy indicators; including a direct comparison of privacy budgets would highlight the practical improvement. The absence of such evaluations leaves a slight gap in quantifying the benefit. Finally, the datasets used are IID or Dirichlet-split with relatively few clients (at most 100) – exploring more heterogeneous or larger-scale federated settings (as done in some FL-DP literature) could further convince readers of the broad applicability. These omissions, however, are not fundamental flaws but areas where the evaluation could be more exhaustive.

---

> ### Author Rebuttal · Authors · 2025-07-31
>
> We sincerely appreciate the positive feedback and are also grateful for the valuable suggestions for improvement. These comments are of great significance for further refining our submission. In this rebuttal discussion, we will address the questions one by one.
>
> ## Q1: How do the contributions of this paper go beyond prior analyses of DP-SGD?
>
> We would like to thank the reviewer for sharing a recent ICLR 2025 work and will cite this work in our submission.
>
> First, we would like to point out that the convergent privacy guarantees established in earlier works for convex and strongly convex functions arise naturally, as their global sensitivity does not diverge. However, this conclusion cannot be directly extended to non-convex functions. Therefore, all the key technical contributions discussed below focus on the non-convex setting.
>
> **Challenges in the analysis from SGD to FL**
>
> We would like to clarify that the analysis from SGD to the FL setting is not straightforward; the final results cannot be derived merely by simply assembling certain existing lemmas or linearly scaling the local training length $K$. In contrast to standard Noisy-SGD, where interpolation can be applied per step, FL requires using a group of specific interpolation coefficient at each aggregation step. This constraint significantly complicates the process of identifying the worst-case privacy. In fact, our early attempts show that a direct extension of interpolation analysis under the local training setting is nearly intractable.
>
> **Existing techniques and our contributions**
>
> As noted by the reviewer, the interpolation technique for SGD under the f-DP framework is not our original contribution; we have properly cited the paper in our manuscript. However, this technique cannot be directly applied to the FL setting. To bridge the gap caused by the heterogeneous nature of multi-node local training in FL, we have further developed auxiliary technique that explicitly measure data and model sensitivities during local training (Figure 1 (right) and Section 4.2 of our paper). Additionally, due to the difficulty of solving the minimax problem for the worst-case privacy loss, we propose a relaxation approach (Section 4.3). Together, these contributions the convergent privacy theory to the FL.
>
> **Our novel findings**
>
> In addition to providing the first convergent privacy guarantee for non-convex FL-DP settings, we also prove that increasing the number of clients also leads to stronger privacy protection, which has not been discussed in previous related work. Moreover, our paper systematically analyzes the impact of local training interval, participating clients, and learning rates. These results challenge some common conclusions in previous works and provide new insights into privacy for FL-DP.
>
>
> ## Q2: Could the authors clarify the impact of this relaxation on the final privacy bound?
>
> We sincerely appreciate the reviewer for raising this highly professional academic question. The validity of our relaxation can be supported by comparison with existing studies, particularly the discussion in Bok et al., 2024 regarding DP-SGD. The core difficulty lies in the fact that, when measuring the worst-case privacy loss, the minimization of the privacy w.r.t. both $t_0$ and ${\lambda_t}$ is a NP-hard problem. To achieve effective estimation, Bok et al. directly assume that the diameter of the entire optimization space is $D$. Therefore, the global sensitivity at any $t_0$ is upper bounded by $D$, which is a constant.
>
> In contrast, our proof avoids making such an assumption. We directly evaluate the outcome of the relaxed formulation. Regarding the reviewer's concern about the tightness of our bound, we would like to clarify that when $m=1$ and $K=1$, FL-DP degrades to DP-SGD, and our results is consistent with the bound presented in Bok et al.
>
> **Why not assume that the global sensitivity at $t_0$ is a constant?**
>
> While this is an optimization trick, this projection is seldom adopted in real-world training. If $t_0$ were an identifiable constant, then this assumption would certainly hold. However, if $t_0$ grows with the total number of steps $T$, the assumption may no longer be valid.
>
> Global sensitivity corresponds to the stability bound in generalization analysis. In the non-convex setting, existing results have not proven that such stability bounds are convergent, and they still diverge as $T$ increases. To avoid introducing additional ambiguity, we chose not to follow the assumption made in Bok et al. (2024). Instead, we directly derive a relaxed upper bound. From our results, this relaxation is effective when $T$ is sufficiently large.
>
> ## Q3: The experimental results are encouraging, but I have suggestions to bolster them.
>
> We sincerely appreciate the reviewer's suggestions regarding the experiments. Due to the sudden change in the NeurIPS rebuttal policy, we are only able to present our experiments in the form of tables. We acknowledge that tabular presentation may be less intuitive, and we would be more than happy to provide additional detailed explanations if there are any further questions.
>
> **More clients**
>
> We increase the number of clients to 500 to measure how the **sensitivity** evolves with respect to time $T$.
>
> |     | T=100 | T=200 | T=300 | T=400 | T=500 | T=600 |
> |:---:|:-----:|:-----:|:-----:|:-----:|:-----:|:-----:|
> |m=50 |  10.62|  16.94|  20.45|  23.13|  23.88|  24.04|
> |m=500|  4.47 |  6.98 |  7.75 |  8.42 |  8.67 |  8.77 |
>
> It can be observed that the convergence trend aligns with our results.
>
> **Results on ViT-Small**
>
> We conducted tests on ViT-Small, and the results show a similar trend to ResNet-18, although the sensitivity is higher, the convergence behavior remains largely consistent.
>
> setup: m=50, K=5, V=10, to ensure satisfactory convergence of the ViT-Small model, we increased the number of training steps to $T=1000$
>
> |     | T=200 | T=400 | T=600 | T=800 | T=1000 |
> |:---:|:-----:|:-----:|:-----:|:-----:|:-----:|
> |ResNet-18 |  16.94|  23.13| 24.04|  24.11|  24.17|
> |ViT-Small|  23.52 |  31.37 |  35.66 |  36.32 |  36.58 |
>
> Sensitivity of ResNet-18 converges after approximately 600 steps, while that of ViT-Small converges after 800 steps.
>
> **Budget $\epsilon$**
>
> We show the theoretical budget $\epsilon$ under the following setup and analyzed its variations under different studies.
>
> Setup: m=50, K=5, V=10, simga=1
>
> |     | T=40 | T=80 | T=120 | T=160 | T=200 |
> |:---:|:-----:|:-----:|:-----:|:-----:|:-----:|
> | [P1] | 0.724 | 1.448 | 2.172 | 2.896 | 3.62 |
> | [P2] | 0.691 | 1.382 | 2.073 | 2.764 | 3.455 |
> | our  | 0.415 | 0.768 | 0.942 | 0.983 | 0.986 |
>
> Due to the limited time during the NeurIPS rebuttal period, we have only been able to complete the experiments presented above. If the reviewers have further suggestions regarding the experiments, we will add more experiments in the final version.
>
> [P1] Differentially private federated learning with local regularization and sparsification. In Proceedings of the IEEE/CVF conference on computer vision and pattern recognition, pages 10122-10131, 2022.
>
> [P2] Dp-norm: Differential privacy primal-dual algorithm for decentralized federated learning. IEEE Transactions on Information Forensics and Security, 2024.
>
> ## Q4: Further analysis and explanation of the key hyperparameters.
>
> We appreciate the reviewer for raising this point. We fully agree that theoretical advancements should aim to provide principled analyses of key hyperparameters. Below, we offer detailed clarifications and additional explanations regarding the specific aspects mentioned by the reviewer.
>
> **Proxy coefficient $\alpha$ in Noisy-FedProx**
>
> Essentially, this is an optimization problem of identifying the largest possible $\alpha$ that still ensures convergence. In fact, proxy terms like this frequently appear in various FL methods. For models such as ResNet, setting $\alpha=0.1 / 0.05$ often yields good convergence behavior. These values are commonly adopted in related works and have also proven to be optimal ranges based on our extensive experimental experience. When the number of clients $m$ increases, $\alpha$ can be slightly increased, but it generally should not exceed $0.5$. For larger models, this term typically needs to decay to $0.01$ or even smaller (e.g., $0.001$) to maintain effective convergence.
>
> **Clipping norm $V$ and local interval $K$ in Noisy-FedAvg**
>
> To enhance privacy, the clipping threshold $V$ should ideally be kept small; however, this can significantly constrain the gradient norm. From an optimization perspective, $V$ serves to suppress extreme fluctuations—such as loss spikes caused by bad data that lead to gradient explosions with norms exceeding $100$. The commonly used setting is $V = 10$, but in practice, this value is often reduced as model size increases. For instance, in LLM training, it is now common to set $V = 1$. This trend not only improves training stability but also benefits privacy protection.
>
> The number of local training steps $K$ plays a key role in reducing the frequency of communication rounds and thereby alleviating communication overhead. However, its impact can be counteracted by the choice of local learning rate, as demonstrated in Table 1 in our paper. When an iteratively decaying learning rate is applied, $K$ no longer affects the worst-case privacy guarantees. In fact, within Noisy-FedProx, even a constant learning rate does not compromise worst-case privacy as long as $\alpha$ satisfies certain conditions. Therefore, the choice of $K$ can be fully driven by optimization considerations, provided that an appropriate learning rate schedule is used.
>
> **We truly believe that engaging in further discussion with you will help us improve our submission. We are very happy to share more insights and clarifications regarding our work, and would be glad to continue the discussion if you have any further questions.**

---

> > ### Comment · Reviewer_eb6Y · 2025-08-03
> >
> > The author's rebuttal address most of my concerns, so I will keep the positive rating score.

---

> > > ### Author Response · Authors · 2025-08-04
> > >
> > > Thank you very much for your valuable feedback in the review. Based on the discussion during the rebuttal phase, we will incorporate the following revisions into the final version, we will add remarks to clarify the assumptions and conditions adopted in previous works for solving this problem, as well as provide a more detailed analysis of our own results; we will supplement explanations for the involved hyperparameters; and we will include the relevant experimental results discussed in the rebuttal. Once again, we sincerely appreciate your time and effort in reviewing our paper. If you have any further questions or suggestions, please feel free to reach out and we will do our best to address your concerns thoroughly.

---

### Note · Authors · 2025-08-13

Dear AC and Reviewers,

We sincerely thank all of you for your valuable time and effort to reviewing our submission, and we are also very grateful to all reviewers for participating in the discussion and acknowledging that our responses have resolved most major concerns.

By the end of the rebuttal discussion, we had not received further questions from the reviewers. Therefore, we have compiled and summarized the issues addressed in the rebuttal and updated them in the new version. Here, we briefly present a summary of the revisions.

    (a) We have added a short paragraph in the section 4 to introduce the differences between DP-SGD and DP-FL analyses and discuss the case when $T/t_0$​ increases. We would like to thank reviewers eb6Y and 6nCw for their insightful suggestions.

    (b) We have added the theoretical results concerning partial participation in remark 3.1 and 4.1. We would like to thank reviewer ByT8 for the constructive suggestion.

    (c) We have added an additional case to Table 1 and Conclusion 4 of DP-FedProx using a decaying learning rate schedule when $\alpha < L$. We thank Reviewer 6nCw for raising this insightful question.

    (d) We have added the motivation for our choice of the base algorithm in the submission. We thank Reviewer 4A4P for the constructive suggestion.

The remaining changes primarily consist of supplementary experiments and additional discussion have also been incorporated into the revision.

**Target of our submission:** We hope this work will provide researchers in the field with new insights into DP-FL analysis. While convergent DP bounds for the classical DP-SGD have made meaningful progress, the FL setting remains insufficiently explored. Since privacy protection is a core principle in the design of FL frameworks, advancing research in this direction is especially important. By exploring how the learning rate, client size, local training interval, and gradient clipping affect the convergent privacy of DP-FL, we can more accurately assess its performance, which also can offer insights for the subsequent algorithm design.

---

### Decision · Program_Chairs · 2025-09-17

**Decision:**

Reject

**Comment:**

This submission considers the application of the shifted interpolation technique, a recently developed method in differential privacy to argue about the privacy of the last iterate, in order to bound the privacy curve of noisy optimization algorithms in Federated Learning.

Reviewers were mildly positive about the work. At the same time multiple reviewers show concerns about the techniques being incremental. Furthermore, I am concerned about the suitability of the model in the context of privacy in distributed settings. Shifted interpolation (and its closely related privacy amplification by iteration) are only applicable to the last released model. In FL the models need to be shared with local devices to compute gradients. In this case, if the devices act maliciously (and are able to collude) they can easily collect all intermediate models: in this context, there is no privacy amplification, and privacy curves may be unbounded.

I couldn't find any discussion around these matters in the paper. Given that such assumptions are crucial for the validity of the conclusions regarding privacy, I recommend this paper to be rejected.